# Distributed Gradient Descent with Many Local Steps in Overparameterized Models

## Abstract

In distributed training of machine learning models, gradient descent with *local iterative steps* is a very popular method, variants of which are commonly known as Local-SGD or the Federated Averaging (FedAvg). In this method, gradient steps based on local datasets are taken independently in distributed compute nodes to update the local models, which are then aggregated intermittently. Although the existing convergence analysis suggests that with heterogeneous data, FedAvg encounters quick performance degradation as the number of local steps increases, it is shown to work quite well in practice, especially in the distributed training of large language models. In this work we try to explain this good performance from a viewpoint of implicit bias in Local Gradient Descent (Local-GD) with a large number of local steps. In overparameterized regime, the gradient descent at each compute node would lead the model to a specific direction locally. We characterize the dynamics of the aggregated global model and compare it to the centralized model trained with all of the data in one place. In particular, we analyze the implicit bias of gradient descent on linear models, for both regression and classification tasks. Our analysis shows that the aggregated global model converges exactly to the centralized model for regression tasks, and converges (in direction) to the same feasible set as centralized model for classification tasks. We further propose a Modified Local-GD with a refined aggregation and theoretically show it converges to the centralized model in direction for linear classification. We empirically verified our theoretical findings in linear models and also conducted experiments on distributed fine-tuning of pretrained neural networks to further apply our theory.

## 1 Introduction

In this era of large machine learning models, distributed training is an essential part of machine learning pipelines. It can happen in a data center with thousands connected compute nodes Sergeev & Del Balso (2018); Huang et al. (2019), or across several data centers and millions of mobile devices in federated learning Konečný et al. (2016); Kairouz et al. (2019). In such a network, the communication cost is usually the bottleneck in the whole system. To alleviate communication burden, and also to preserve privacy to some extent, one common strategy is to perform multiple local updates before sending the information to other nodes, which is called Local Gradient Descent (Local-GD) Stich (2019); Lin et al. (2019). It is also a standard algorithm in federated learning, varied by partial device participation and privacy constraints, and known as FedAvg McMahan et al. (2017). While local updates can reduce communication cost, the number of local steps is usually considered to be small Stich (2019); Li et al. (2020b). When data distributions across machines are heterogeneous, a large number of local steps would result in local iterates to diverge significantly (called client-drift), and the aggregated values to oscillate and be far away from the optimum global model.

However, in practical implementation of distributed training on large models, the performance of vanilla FedAvg is surprisingly good even with heterogeneous data distribution McMahan et al. (2017); Charles et al. (2021). In fact SCAFFOLD Karimireddy et al. (2020), an algorithm designed to mitigate the effect of heterogeneity theoretically, is shown to have similar empirical performance as FedAvg Reddi et al. (2021); Wu et al. (2023). There are some works trying to explain the effectiveness of FedAvg from different theoretical aspects, such as representation learning Collins et al. (2022), refined theoretical assumption Wang et al. (2024) etc. Also, the number of local steps can be very large in real-world systems, for example, performing 500 local steps in distributed training of large language

models (LLM) Douillard et al. (2023); Jaghouar et al. (2024). These practical experiences motivates us to consider the following question:

*Q: Can we establish rigorous conditions, independent of data distribution, under which Local-GD performs well with a very large number of local steps?*

In this work we answer this question in affirmative by considering overparameterized models on regression and classification tasks. Our main tool is to analyze the *implicit bias* of gradient descent to characterize the dynamics of aggregated models with many local steps. In a network with $M$ compute nodes, the goal is to train a global model to fit in the distributed datasets:

$$\min_{w\in\mathbb{R}^d} f(w) \qquad \text{with } f(w) \equiv \frac{1}{M}\sum_{i=1}^{M} f_i(w|D_i), \tag{1}$$

where $w \in \mathbb{R}^d$ is the single model to be trained and $f_i(w|D_i)$ is the local objective function, and $D_i$ is the local distribution of $d$-dimensional samples and corresponding labels $\{x_{ij}, y_{ij}\}_{j=1}^{N}$.

To reduce the communication frequency, Local-GD chooses to do $L$ local gradient descent steps before sending the local model to a central node. The detailed algorithm of Local-GD is described in Algorithm 1 and 2. In the existing convergence analysis of Local-GD, the number of local steps $L$ should not be very large. For example, with strongly convex and smooth loss functions, the number of local steps should not be larger than $O(\sqrt{T})$ for i.i.d data Stich (2019) and non-i.i.d. data Li et al. (2020b). However, such analysis is developed for general/classical models and does not consider the special properties of overparameterized models. In this work we specifically focus on linear models for both regression and classification tasks and take the overparameterized regime into account. That is, the dimension $d$ is larger than the total number of samples, i.e. $d > MN$. While modern machine learning concerns primarily large nonlinear models, it is instructive to explore the intrinsic property of Local-GD in simpler linear setting and establish the connection to other areas. For example, the leading theories of deep learning, such as implicit bias of optimization algorithms, or double descent Belkin et al. (2018; 2019), were built for linear models first. Moreover, fine-tuning on pretrained large models has gradually become the popular paradigm in practical machine learning pipeline. It is widely used to fine-tune the final linear layer or add a few linear layers to pretrained models in transfer learning Donahue et al. (2014); Kornblith et al. (2019) and deployment of LLM Devlin (2018); Jiang et al. (2020).

As stated, to characterize the behavior of Local-GD with large number of local steps in overparameterized models, we leverage the implicit bias of gradient descent, which is an active area in theoretical explanation of modern large models Soudry et al. (2018); Gunasekar et al. (2018a); Ji & Telgarsky (2019a); Chizat & Bach (2020); Frei et al. (2024). With a very large number of local steps, the local optimization problem can be exactly solved for linear regression and classification models. In overparameterized regime, gradient descent would converge to a specific solution. After aggregation of these specific local solutions, we can characterize the dynamic of the global model and finally compare it to the centralized model trained on a collection of distributed datasets at one place.

Specifically, in linear regression minimizing a squared loss, the local models would fit to the corresponding local datasets, and converge to the solution with minimum distance to initial aggregated global model at each communication round. We can obtain the closed form of this solution and calculate the global model after aggregation. We prove that it exactly converges to the centralized model (the model trained by gradient descent if all data were in one place) as the number of rounds of communication increases.

The analysis of linear classification (halfspace learning) is more involved and proceeds according to the following steps. First, it turns out that when minimizing an exponential loss with a weakly regularized term, the aggregated global model is equivalent to a model aggregated from local models obtained by solving *local max-margin* problems. Subsequently we relate the update of global model aggregated from solutions of local max-margin problems to *Parallel Projection Method (PPM)*, an iterative algorithm used for finding a point in the intersection of multiple constraint sets by projecting onto each constraint set in parallel Gilbert (1972); de Pierro & Iusem (1984); Combettes (1994; 1996). Using properties of PPM, we can characterize the dynamics of the aggregated global model. We prove that it converges to a global feasible set, which is the intersection of constraint sets in local max-margin problems. The centralized model trained with all of the data also converges to the global feasible set. To further explain the similar performance obtained by global model and centralized model, we propose a *modified* Local-GD with a different aggregation method from vanilla Local-GD (Algorithm 3). We theoretically prove that the aggregated global model obtained from Modified Local-GD exactly

converges to the centralized model in direction. We show the vanilla Local-GD actually converges to the same point as the modified Local-GD experimentally. For both linear regression and classification, our results show that the aggregated global model would converge to the centralized model even with a very large number of local steps on heterogeneous data.

In summary, the contribution of this work is as follows:

- We established the theoretical performance of Local-GD with a large number of local steps in overparameterized models. We analyzed the implicit bias of Local-GD, for single communication round of linear regression, and for whole algorithmic process of classification, respectively. As far as we know, this is the first attempt to analyze implicit bias of gradient descent in distributed setting.

- We obtained closed form of the aggregated global model in linear regression and analyzed its dynamics. We proved that it exactly converges to the centralized model as communication rounds increase.

- We related the Local-GD for linear classification to Parallel Projection Method and characterized the dynamics based on the properties of projections. We proved the aggregated global model converges to a global feasible set same as the centralized model.

- We further proposed a Modified Local-GD with a different aggregation method and proved it converges exactly to the centralized model in direction.

- We experimentally verify our theoretical findings on synthetic datasets and real datasets with linear models. We further conducted experiments on fine-tuning the final linear layer of neural networks to show the broader impact of our work.

Our main technical challenge comes while analyzing classification. In linear regression, the implicit bias for a single round of communication is directly derived from the gradient on squared loss (each gradient step is on the row space of local data). In contrast, for classification we have to consider the whole algorithmic process of both Local-GD and Parallel Projection Method and then derive the equivalence between them. Compared to the continual learning work Evron et al. (2023) where overparameterized models are handled sequentially, the challenge is that we need to handle the parallel projections happening *simultaneously* from the same initial point. Due to space limit, we give more additional references and discussion on Related Works in Appendix A.

---

**Algorithm 1** LOCAL-GD.

---

1: **Input:** learning rate $\eta$.
2: Initialize $w_0^0$
3: **for** $k = 0$ to $K - 1$ **do**
4:     The aggregator sends global model $w_0^k$ to all compute nodes.
5:     **for** $i = 1$ to $i = M$ **do**
6:         compute node $i$ updates local model starting from $w_0^k$: $w_i^{k+1} = \mathsf{LocalUpdate}(w_0^k)$.
7:         compute node $i$ sends back the updated local model $w_i^{k+1}$.
8:     **end for**
9:     The aggregator aggregates all the local models: $w_0^{k+1} = \frac{1}{M} \sum_{i=1}^{M} w_i^{k+1}$.
10: **end for**
11: **Output:** $w_0^K$.

---

---

**Algorithm 2** $\mathsf{LocalUpdate}(w_0^k)$ in general Local-GD.

---

1: **Input:** an initial point $w_0^k$, the number of local steps $L$, and the learning rate $\eta$.
2: Initialize $w_i^{k,0} = w_0^k$.
3: **for** $l = 0$ to $L - 1$ **do**
4:     $w_i^{k,l+1} = w_i^{k,l} - \eta \nabla f_i(w_i^{k,l})$.
5: **end for**
6: **Output:** $\mathsf{LocalUpdate}(w_0^k) := w_i^{k,L}$.

---

## 2 LOCAL-GD IN LINEAR REGRESSION: A WARM-UP

### 2.1 SETTING

In this section we first consider linear regression in overparameterized regime. The behavior of linear regression is very well-understood in high-dimensional statistics; and we can clearly convey our key message based on this fundamental setting.

At each compute node $i$, the dataset $S_i$ consists of $N$ tuples of samples and their corresponding labels, $(x, y) \in \mathbb{R}^d \times \mathbb{R}$. We assume the label $y_{ij}$ is generated by

$$y_{ij} = x_{ij}^T w_i^* + z_{ij} \tag{2}$$

where $w_i^* \in \mathbb{R}^d$ is the ground truth model at $i$-th compute node, and $z_{ij}$ is the added noise. Denote $X_i = [x_{i1}, x_{i2}, ..., x_{iN}]^T \in \mathbb{R}^{N \times d}$ as the data matrix at $i$-th compute node, and $y_i = [y_{i1}, y_{i2}, ..., y_{iN}] \in \mathbb{R}^N$ as the label vector, $z_i \in \mathbb{R}^N$ as the noise vector. In heterogeneous setting, the $w_i^*$ can be very different to each other. Note that the convergence to centralized model does not rely on the generative model. We just make this assumption on generative model for deriving a more clear form of the aggregated global model.

**Algorithm.** At each round, the aggregator sends the global model $w_0$ to all the compute nodes. Each compute node minimizes the squared loss $f_i(w_i) = \frac{1}{2N}\|y_i - X_i w_i\|^2$ by a large number of gradient descent steps *until convergence*. Then each compute node sends back the local model and the aggregator aggregates all the local models to get the updated global model. The detailed algorithm is Local-GD in Algorithm 1 with $f_i(w_i)$ replaced in LocalUpdate (Algorithm 2). Since minimizing squared loss is a quadratic problem, it is expected to reach convergence locally with a small number of gradient descent steps.

### 2.2 IMPLICIT BIAS OF LOCAL GD IN LINEAR REGRESSION

For each local problem, when the dimension of the model is larger than the number of samples at each compute node ($d > N$), i.e., locally overparameterized, there are multiple solutions corresponding to zero squared loss. However, gradient descent will lead the model converge to a specific solution, which corresponds to a minimum Euclidean distance to the initial point Gunasekar et al. (2018a); Evron et al. (2022). Formally, the solution $w_i^{k+1}$ obtained at $k$-th round and $i$-th node will converge to the solution of the optimization problem

$$\min_{w_i} \quad \|w_i - w_0^k\|^2 \quad \text{s.t.} \quad X_i w_i = y_i. \tag{3}$$

We can obtained the closed form solution of this optimization problem as (see Proof of Lemma 1 in Appendix C.1)

$$w_i^{k+1} = \left(I - X_i^T (X_i X_i^T)^{-1} X_i\right) w_0^k + X_i^T (X_i X_i^T)^{-1} y_i \tag{4}$$
$$= \left(I - X_i^T (X_i X_i^T)^{-1} X_i\right) w_0^k + X_i^T (X_i X_i^T)^{-1} X_i w_i^* + X_i^T (X_i X_i^T)^{-1} z_i.$$

Denote $P_i \triangleq X_i^T (X_i X_i^T)^{-1} X_i$ and $X_i^\dagger \triangleq X_i^T (X_i X_i^T)^{-1}$. The local model can be rewritten as $w_i^{k+1} = (I - P_i) w_0^k + P_i w_i^* + X_i^\dagger z_i$. We observe that $P_i$ is the projection operator to the row space of $X_i$, and $X_i^\dagger$ is the pseudo inverse of $X_i$. After one round of iterations, the local model is actually an interpolation between the initial global model $w_0^k$ at this round and the ground-truth model $w_i^*$, plus a noise term. We then obtain the closed form of global model by aggregation. After many rounds of communication, we can obtain the final trained global model from Local-GD.

**Lemma 1.** *When the local overparameterized linear regression problems are exactly solved by gradient descent, then after $K$ rounds of communication, the global model $w_0^K$ obtained from Local-GD is*

$$w_0^K = (I - \bar{P})^K w_0^0 + \sum_{k=0}^{K-1} (I - \bar{P})^k (\bar{Q} + \bar{Z}), \tag{5}$$

*where $\bar{P} = \frac{1}{M}\sum_{i=1}^M P_i, \bar{Q} = \frac{1}{M}\sum_{i=1}^M P_i w_i^*, \bar{Z} = \frac{1}{M}\sum_{i=1}^M X_i^\dagger z_i.$*

Note that $\bar{P}, \bar{Q}, \bar{Z}$ are constant after the data is generated. Since we only know the $\{X_i, y_i\}_{i=1}^M$ in the training process, we can also write it as

$$w_0^K = (I - \bar{P})^K w_0^0 + \sum_{k=0}^{K-1} (I - \bar{P})^k \bar{Y}, \tag{6}$$

where $\bar{Y} = \frac{1}{M} \sum_{i=1}^M X_i^\dagger y_i$. Then we can directly get the final model from the training set.

**Singularity of $\bar{P}$.** If $\bar{P}$ is invertible, we can further simplify the form of global model. However, since $P_i \in \mathbb{R}^{d \times d}$ is the projection operator onto row space of $X_i$, its rank is at most N. The $\bar{P}$ is the average of $P_i$s, thus its rank is at most $MN$. Note that we consider the overparameterized regime both locally and globally, i.e., $d \gg MN$. Then $\bar{P}$ is singular, and the sum $\sum_{k=0}^{K-1} (I - \bar{P})^k$ approaches $KI$ when $d$ becomes very large. We cannot get more properties of the final global model from (6), but we can compare it to the centralized model trained with all of the data.

## 2.3 CONVERGENCE TO CENTRALIZED MODEL

Let $X_c = [X_1^T, \dots, X_M^T]^T \in \mathbb{R}^{MN \times d}$ be the data matrix consisting of all the local data, and $y_c = [y_1^T, \dots, y_M^T]^T \in \mathbb{R}^{MN \times 1}$ be the label vector consisting of the local labels. If we train the centralized model from initial point 0 with squared loss, then the gradient descent will lead the model to the solution of the optimization problem

$$\min_w \quad \|w\|^2 \quad \text{s.t.} \quad X_c w = y_c \tag{7}$$

We can write the closed form of centralized model as $w_c = X_c^T (X_c X_c^T)^{-1} y_c$.

Due to the constraint in problem (7), for each compute node $i$, we have $X_i w_c = y_i$. We replace $y_i$ in the local model (4), then we have

$$w_i^{k+1} - w_c = (I - P_i)(w_0^k - w_c). \tag{8}$$

The right-hand side is projecting the difference between global model and centralized model onto null space of $X_i$. After averaging all the local models at the aggregator, we have

$$w_0^{k+1} - w_c = (I - \bar{P})(w_0^k - w_c). \tag{9}$$

In the training process the difference between global model and centralized model is iteratively projected onto the null space of span of row spaces of $X_i$s. It implies that the difference on the span of data matrix gradually decreases until zero. Based on the evolution of the difference, we can prove the following theorem:

**Theorem 1.** *For the linear regression problem, suppose the initial point $w_0^0$ is 0 and $d \gg MN$, then the global model obtained by Local-GD, $w_0^K$, converges to the centralized solution $w_c$ as the number of communication rounds $K \to \infty$.*

The proof is deferred in Appendix C.2. The key step is to show the initial difference is already in the data space, and no residual in the null space of row spaces of $X_i$s.

Due to the linearity of the regression problem, we can theoretically show the global model can exactly converge to the centralized model with implicit bias on overparameterized regime. Note that the proof does not rely on the generative model and assumption on data heterogeneity. It implies that, even if we use a large number of local steps to exactly solve the local problems on very heterogeneous data, the performance of Local-GD is equivalent to train a model with all the data in one place.

# 3 LOCAL-GD IN LINEAR CLASSIFICATION: RELATION TO PPM

## 3.1 SETTING

In this section we investigate a binary classification task with linear models. Different from the linear regression problem, it is hard to obtain closed form solution on classification tasks. Thus we need to develop new techniques to handle this case.

Suppose, for each compute node $i$, the dataset $S_i$ consists of $N$ tuples of samples and their corresponding labels, $(x, y) \in \mathbb{R}^d \times \{+1, -1\}$. Similarly, we denote $X_i \in \mathbb{R}^{N \times d}$ as the data matrix at $i$-th compute node, and $y_i \in \{+1, -1\}^N$ as the label vector. We do not assume the generative model in classification task, but we need an assumption of separable datasets.

**Assumption 1.** Each local dataset $S_i$ is separable, i.e., there are non-empty local feasible sets,

$$C_i \triangleq \{w \in \mathbb{R}^d \mid y_{ij} x_{ij}^T w \geq 1, \text{for } j = 1, ..., N\}, \tag{10}$$

and there is a non-empty global feasible set,

$$\bar{C} \triangleq \cap_{i=1}^m C_i \neq \emptyset. \tag{11}$$

This assumption makes sure that the datasets are locally and globally separable.

**Algorithm.** At each round, the aggregator sends the global model $w_0$ to all the compute nodes. Each compute node minimizes an exponential loss with a weakly regularized term by many gradient descent steps *until convergence*. That is, each compute node solves the following problem:

$$\min_{w \in \mathbb{R}^d} f_i(w_i) = \sum_{j=1}^N \exp\left(-y_{ij} x_{ij}^T w\right) + \frac{\lambda}{2} \|w - w_0^k\|^2 \tag{12}$$

where $\lambda$ is a regularization parameter close to 0.

Then each compute node sends back the local model and the aggregator aggregates all the local models to get the updated global model. The detailed algorithm for linear classification is Local-GD in Algorithm 1 with $f_i(w_i)$ replaced in LocalUpdate (Algorithm 2).

Regularization methods are very common in distributed learning to force the local models move not too far from global model Li et al. (2020a; 2021); T Dinh et al. (2020). Here we consider the weakly regularized term, $\lambda \to 0$, to give theoretical insights of Local-GD on classification tasks. Experimentally the $\lambda$ is set to be extremely small that does not affect the minimization of exponential loss. Since the local problem is a strongly convex problem, with many local gradient descent steps it will be exactly solved.

## 3.2 IMPLICIT BIAS OF GRADIENT DESCENT IN LINEAR CLASSIFICATION

One can derive the implicit bias of classification at a single local node after a large number of local steps. However, in contrast to linear regression, we cannot easily aggregate the local solutions after a round of communication to a closed form. At each round, the local model is updated from the previously aggregated global model, which is related to previous local updates. To mitigate this, we consider the whole algorithmic process of Local-GD on classification and use another auxiliary sequence of global models, denoted as $\bar{w}_0^k, k = 0, 1, 2, ....$ Starting from an initial point $\bar{w}_0^0$, the central node sends global model $\bar{w}_0^k$ to all the compute nodes at $k$-th iteration round. Each compute node solves the following Local Max-Margin problem to obtain $\bar{w}_i^{k+1}$:

$$\bar{w}_i^{k+1} = \arg\min_{w \in \mathbb{R}^d} \|w - \bar{w}_0^k\| \quad \text{s.t.} \quad y_{ij} x_{ij}^T w \geq 1 \quad j = 1, 2, ..., N. \tag{13}$$

Then the compute node sends the local model back. The central node averages the local models to get $\bar{w}_0^{k+1} = \frac{1}{M} \sum_{i=1}^M \bar{w}_i^{k+1}$.

We can show the solution $w_0^K$ obtained in Local-GD converges in direction to the global model from Local Max-Margin problems $\bar{w}_0^K$.

**Lemma 2.** *For almost all datasets sampled from a continuous distribution satisfying Assumption 1, with initialization $w_0^0 = \bar{w}_0^0 = 0$, we have $w_0^k \to \ln\left(\frac{1}{\lambda}\right) \bar{w}_0^k$, and the residual $\|w_0^k - \ln\left(\frac{1}{\lambda}\right) \bar{w}_0^k\| = O(k \ln\ln \frac{1}{\lambda})$, as $\lambda \to 0$. It implies that at any round $k = o\left(\frac{\ln(1/\lambda)}{\ln\ln(1/\lambda)}\right)$, $w_0^k$ converges in direction to $\bar{w}_0^k$:*

$$\lim_{\lambda \to 0} \frac{w_0^k}{\|w_0^k\|} = \frac{\bar{w}_0^k}{\|\bar{w}_0^k\|}. \tag{14}$$

The proof is deferred in Appendix D. The framework is similar to the continual learning work Evron et al. (2023), but we need to handle the parallel local updates for each dataset from the same initial

model and the aggregation, which is different from the sequential updates where for each dataset the model is trained from the previous model and there is no need to do aggregation.

Based on this equivalence between Local-GD for linear classification and Local Max-Margin scheme, we can further analyze the performance of Local-GD with a large number of local steps. Instead of a closed-form solution for the Local Max-Margin problem (13), we treat it as a projection of the aggregated global model onto a convex set $C_i$: $\bar{w}_i^{k+1} = P_i(\bar{w}_0^k)$, which is formed by the constraints in (13) and exactly the local feasible set defined in Assumption 1. Here we slightly overload the notation $P_i$, which was used as the projection matrix in linear regression since the readers can get a sense of the same effect of them in Local-GD. The aggregation is actually to average the local projected points: $\bar{w}_0^{k+1} = \frac{1}{M}\sum_{i=1}^{M} P_i(\bar{w}_0^k)$.

The sequence of Local Max-Margin schemes is therefore projections to local (convex) feasible sets followed by aggregation, which is the Parallel Projection Method (PPM) in literature Gilbert (1972); Combettes (1994). Using Lemma 2, we establish the relation between Local-GD and PPM: the model from Local-GD converges to the model from PPM in direction.

### 3.3 Convergence to Global Feasible Set

Now we use the properties of PPM to characterize the performance of Local-GD in classification. In Combettes (1994), the convergence of PPM has been provided for a relaxed version. The direct average considered in this work can be seen as a special case of the relaxed version, and the following lemma holds.

**Lemma 3** (Theorem 1 and Proposition 8, Combettes (1994))**.** *Suppose all the local feasible sets $C_i, i = 1, 2, \ldots$ are closed and convex, and the intersection $\bar{C}$ is not empty. Then for any initial point $\bar{w}_0^0$, the global model $\bar{w}_0$ generated by PPM converges to a point in the global feasible set $\bar{C}$.*

This lemma guarantees that $\bar{w}_0^K$ will converge to the intersection of the convex sets after many rounds of iteration, however we are not sure which exact point it would converge to.

Similar to linear regression case, we also compare the global model obtained from Local-GD to the centralized model trained with all of data in one place. From the implicit bias of gradient descent on exponential-tailed loss Soudry et al. (2018), the centralized model trained with exponential loss will converge in direction to the solution of a Max-Margin problem:

$$\min_{w \in \mathbb{R}^d} \|w\| \quad \text{s.t.} \quad y_{ij} x_{ij}^T w \geq 1, \quad i = 1, 2, \ldots, M, \quad j = 1, 2, \ldots, N. \tag{15}$$

This problem is actually the problem of hard margin support vector machine (SVM). The constraints in equation 15 include all the local datasets, and form the global feasible set $\bar{C}$. That is, the centralized model would converge to the minimum norm solution in global feasible set in direction.

Combining Lemma 2, Lemma 3 and result of centralized model, we immediately have:

**Theorem 2.** *For linear classification problem with exponential loss, suppose initial point is $w_0^0 = 0$. The aggregated global model $w_0^K$ obtained by Local-GD with a large number of local steps converges in direction to one point in the global feasible set $\bar{C}$, while the centralized model converges in direction to the minimum norm point in the same set.*

The main difference from linear regression is that we cannot guarantee the global model obtained by Local-GD to converge exactly to the centralized model in classification, but show that it converges to the same global feasible set as the centralized solution. Nevertheless, in experiments the test accuracy of the Local-GD model is very similar to that of centralized model. To theoretically support that the Local-GD model converges to the centralized model, we propose a slightly Modified Local-GD by just changing the aggregation method, and showing that it converges to the centralized model exactly.

## 4 Modified Local-GD: Convergence to Centralized Model

Previously, we established the connection between Local-GD and PPM in linear classification. In Combettes (1996) it was shown that if the aggregation method is modified to incorporate the influence of the initial point $\bar{w}_0^0$ in PPM, then the sequence generated by PPM will converge to a specific point

in global feasible set $\bar{C}$ with minimum distance to this initial point. Denote $P_c(\cdot)$ as the projection operator onto the global feasible set $\bar{C}$. Formally we have the following lemma.

**Lemma 4** (Theorem 5.3, Combettes (1996)). *Suppose $\bar{C}$ is not empty. For any initial point $\bar{w}_0^0$, when the local models are aggregated as*

$$\bar{w}_0^{k+1} = (1-\alpha^{k+1})\bar{w}_0^0 + \alpha^{k+1}\left(\frac{1}{M}\sum_{i=1}^{M}P_i(\bar{w}_0^k)\right), \tag{16}$$

*where $\{\alpha^k\}$ satisfy $(i)\lim_{k\to\infty}\alpha^k=1, (ii)\sum_{k\geq 0}(1-\alpha^k)=\infty, (iii)\sum_{k\geq 0}|\alpha^{k+1}-\alpha^k|<\infty$, then the global model generated by PPM will converge to the point $P_c(\bar{w}_0^0)$.*

That is the sequence generated by PPM would converge to the point in global feasible set, $\bar{C}$, with minimum distance to $\bar{w}_0^0$. The modified aggregation method is a linear combination of initial point and current average of local projected points. One example of the sequence $\{\alpha^k\}$ satisfying the conditions is $\alpha^k = 1 - \frac{1}{k+1}$.

If we start from $\bar{w}_0^0 = 0$, then the point $P_c(\bar{w}_0^0)$ is exactly the minimum norm point in the global feasible set. It shows the PPM can exactly converge to the minimum norm point as the centralized model. Based on this result, we propose a Modified Local-GD algorithm shown in Algorithm 3, which only differs from Local-GD in the aggregation method.

---

**Algorithm 3** MODIFIED LOCAL-GD.

---

1: **Input:** learning rate $\eta$.
2: Initialize $w_0^0$
3: **for** $k=0$ to $K-1$ **do**
4:     The central node sends global model $w_0^k$ to all compute nodes.
5:     **for** $i=1$ to $i=M$ **do**
6:         compute node $i$ updates local model starting from $w_0^k$: $w_i^{k+1} = \mathsf{LocalUpdate}(w_0^k)$.
7:         compute node $i$ sends back the updated local model $w_0^{k+1}$.
8:     **end for**
9:     The central node aggregates all the local models: $w_0^{k+1} = (1-\alpha^k)w_0^0 + \alpha^k\left(\frac{1}{M}\sum_{i=1}^{M}w_i^k\right)$.
10: **end for**
11: **Output:** $w_0^K$.

---

We still need to prove a lemma analogous to Lemma 2 to establish the equivalence between Modified Local-GD and Modified PPM, which is omitted here due to space limit (Please refer to Appendix E and the proof is very similar to proof in Lemma 2). From the equivalence, Lemma 4, and result of the centralized model, we can have the following theorem:

**Theorem 3.** *For linear classification problem, suppose the initial point is $w_0^0 = 0$. Then the global model $w_0^K$ obtained by Modified Local-GD (Algorithm 3) converges in direction to the centralized model obtained from (15).*

Unlike the vanilla Local-GD, which is only guaranteed to converge to the global feasible set, the Modified Local-GD is guaranteed to converge to the centralized model in direction. Unlike linear regression, the convergence is established *in direction* since the solution on exponential loss could go to infinity.

Note that if we start from $\bar{w}_0^0 = 0$, the aggregation in Modified Local-GD becomes $w_0^{k+1} = \frac{k}{k+1}\left(\frac{1}{M}\sum_{i=1}^{M}w_i^k\right)$, which is just a *scaling* of vanilla aggregation with a parameter less than 1. Thus we can see experimentally they usually converge to the same point and Modified Local-GD converges slightly slower. In summary, Modified Local-GD theoretically illustrates that the global model trained from Local-GD could obtain similar performance as the centralized model.

## 5 EXPERIMENTS

**Linear Regression.** We simulated 10 compute nodes, each with 50 training samples. The label vector $y_i$ at $i$-th compute node is exactly generated as (2), where ground truth model $w_i^*$ is Gaussian

vector with each element following $\mathcal{N}(0,4)$. Each ground truth model at different compute nodes is independently generated, thus the datasets can be very different from each other. The data matrix $X_i$ also follows Gaussian distribution, with each element being $\mathcal{N}(0,1)$, and $z_i$ is a Gaussian vector with $\mathcal{N}(0,0.04)$. In Local-GD, the number of local steps is $L=200$, number of rounds is also $R=200$, and the learning rate $\eta=0.0001$. Actually it just take a few local steps to converge locally at each round, but we set a large number of local steps to show it can be large at $O(\sqrt{T})$, where $T=L*R$ is the number of total iterations. We tested the global model (G) from Local-GD on squared loss, centralized model (C) trained from global dataset on squared loss, closed form of global model (G-Closed) in (6), closed form of centralized model (C-Closed) as solution of problem (7). The centralized model is trained 10000 steps with learning rate 0.0001.

Fig. 1(a) displays the difference between global model and trained centralized model, and difference between global model and closed form of global model at each round when dimension is $d=1500$, which is locally and globally overparameterized. The difference between two models is $\|w_1-w_2\|/d$. We can see the difference between global model and its closed form is always 0 during the training process, verifying the correctness of the derived closed form (6). The global model can gradually converge to the centralized model with more communication rounds.

Fig. 1(b) displays the difference between global model and centralized model, global model and its closed form, and centralized model and its closed form, with respect to model dimension. Since it is always locally overparameterized, the difference between global model and the closed form is always zero. The difference between global model and centralized model has an obvious peak around 500, which is the number of total samples. The phenomenon that global model converges exactly to centralized model only happens when the model is sufficiently overparameterized. Fig. 1(c) shows the generalization error of global model and centralized model in linear regression. Since the data matrix is Gaussian, the generalization error of model $w$ can be computed as $\frac{1}{M}\sum_{i=1}^{M}\|w-w_i^*\|^2$. We plot the generalization error divided by $d$. It is shown the global model and centralized model can get the same performance when model is sufficiently overparameterized.

**Classification.** For linear classification, we also have 10 compute nodes, with 50 samples at each. The dataset is generated as $y_{ij}=\text{sign}(x_{ij}^T w_i^*)$, where ground truth model is $w_i^*=w^*+z_i$, and $w^*$ is a Gaussian vector randomly chosen, $z_i$ is a Gaussian noise. The data matrix $X_i$ is still a Gaussian matrix. This setting makes sure the datasets across compute nodes are different from each other, meanwhile they are not totally different such that there may be a non-empty global feasible set. The global model is trained exactly as Local-GD for linear classification, where the $\lambda$ is 0.0001. Actually we can use the standard logistic regression without regularization to obtain the same performance. But aligning with theoretical proof, we still use exponential loss with a very weak regularization. We tested global model (G), global model from Modified Local-GD (G-Mod), centralized model (C) from minimizing exponential loss on all the data, centralized SVM model (S) solved from problem (15) via standard scikit-learn package. Note that centralized model and SVM model are the final trained model in the plots. In Local-GD, the number of local steps is $L=150$, the number of communication rounds is $R=120$, and the learning rate is $\eta=0.01$. The centralized model is trained with same learning rate for 20000 steps. Since our theory claimed the convergence is established in direction, the difference computed here for two models is defined after normalization $\|w_1/\|w_1\|-w_2/\|w_2\|\|$.

Fig. 1(d) shows the difference between these models with respect to the number of rounds $R$ when dimension is $d=1500$. We can see both global model and modified global model converges to the centralized model in direction, and the centralized model is close to the SVM model but there is small gap. Fig. 1(e) displays the difference with respect to dimension $d$. It is seen the difference between global model and centralized model gradually decreases with larger dimensions. The modified global model is almost the same as the centralized model but the gap is slightly larger since it converges slower than vanilla global model with same number of rounds. Fig. 1(f) shows the difference from SVM model with dimension. The gap between the models to SVM model also decreases with larger $d$. Finally Fig. 1(g) plots the test accuracy of these models. The test datasets are also constructed by the same generation of training set with different data matrix. Although the accuracy decreases with larger dimension (relatively fewer samples), the performance of global models and centralized models are always similar.

**Fine-Tuning of Pretrained Neural Network.** We further fine-tuned the ResNet50 model pretrained with ImageNet dataset on CIFAR10 dataset. Only the final linear layer is trained during the process, while the rest of model is fixed. The 50000 samples are distributed on 10 compute nodes. For $i$-th

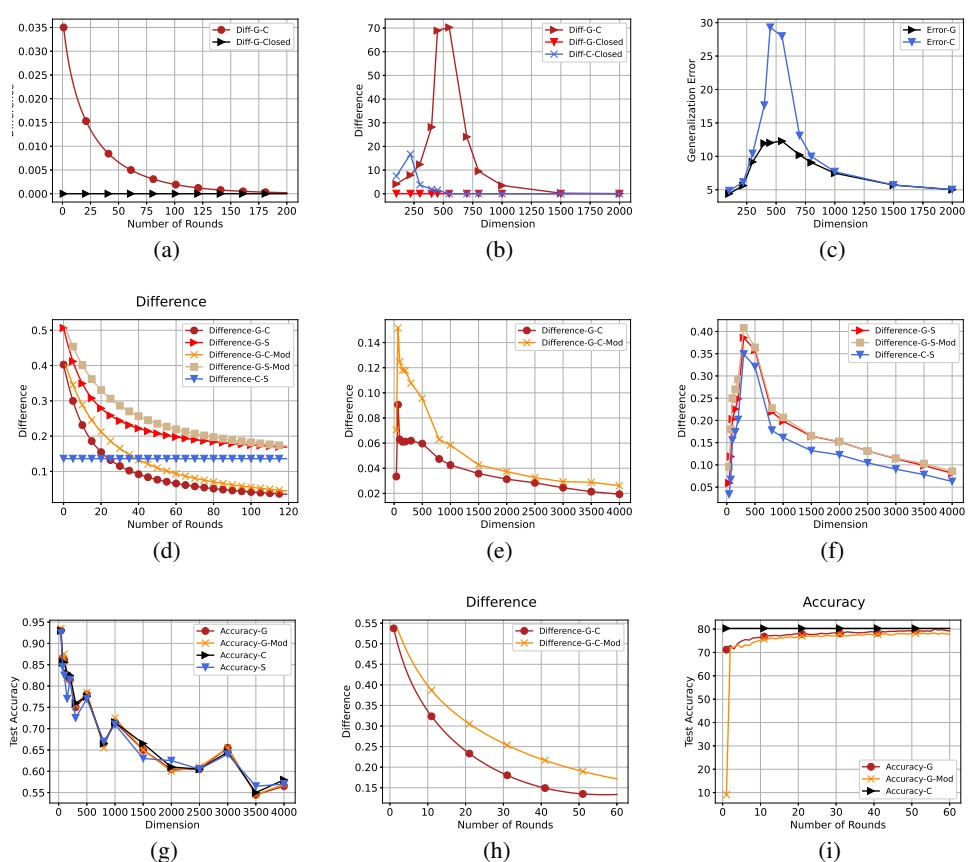

Figure 1: From left to right, from up to bottom (LR: Linear Regression, LC: Linear Classification, NN: Neural Network): (a) Difference between models with communication rounds in LR. (b) Difference between models with dimension in LR. (c) Generalization error with dimension in LR. (d) Difference between global model and centralized model with $R$ in LC. (e) Difference between global model and centralized model with $d$ in LC. (f) Difference from SVM model with $d$ in LC. (g) Test Accuracy in LC. (h) Difference between global model and centralized model with communication rounds in NN. (i) Test accuracy with communication rounds in NN.

compute node, the half of local dataset belongs to the same class, and the other half consists of rest of 9 classes evenly, which forms a heterogeneous data distribution. The centralized model is trained with the whole CIFAR10 dataset. The models are trained with cross entropy loss and SGD. The learning rate is 0.01 and the batch size is 128. The number of local steps is $L = 60$ and number of communication rounds is $R = 60$. The centralized model is trained with the same learning rate for 3600 steps. We plot the difference between the linear layer and test accuracy with number of rounds in Fig. 1 (h) and (i). Again the difference is defined in direction. We can see the difference gradually decreases to a small error floor and the accuracy of global models and centralized model is very similar at last.

## 6 CONCLUSIONS

In this work we analyzed the implicit bias in distributed setting, and characterized the dynamics of global model trained from Local-GD with many local steps based on the implicit bias. We showed that the global model can converge to centralized model for both linear regression and classification tasks, providing a new perspective why Local-GD (FedAvg) works well in practice even with a large number of local steps on heterogeneous data. One potential future work is to extend the analysis of Local-GD to neural network using the developed implicit bias of deeper models Chizat & Bach (2020); Gunasekar et al. (2018b); Ji & Telgarsky (2019b); Kou et al. (2024).

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

# A   RELATED WORK

**Convergence of Local-GD.** When data distribution is homogeneous, many works have been done to establish convergence analysis for Local (Stochastic) GD Stich (2019); Yu et al. (2019); Khaled et al. (2020). With a "properly" small number of local steps, the dominating convergence rate is not affected. Further various assumptions have been made to handle data heterogeneity and develop convergence analysis Li et al. (2020b); Karimireddy et al. (2020); Khaled et al. (2020); Reddi et al. (2021); Wang et al. (2020); Crawshaw et al. (2023). For strongly convex and smooth loss functions, the number of local steps should not be larger than $O(\sqrt{T})$ for i.i.d data Stich (2019) and non-i.i.d. data Li et al. (2020b). However, in practice Local-GD (FedAvg) works well in many applications McMahan et al. (2017); Charles et al. (2021), even in training large language models Douillard et al. (2023); Jaghouar et al. (2024). In Wang et al. (2024), the authors argue that the previous theoretical assumption does not align with practice and proposed a client consensus hypothesis to explain the effectiveness of FedAvg in heterogeneous data. But they do not consider the impact of overparameterization on distributed training. There are some works incorporating the property of zero training loss of overparameterized neural networks into the conventional convergence analysis of FedAvg Huang et al. (2021); Deng et al. (2022); Song et al. (2023); Qin et al. (2022). However, they do not guarantee which point FedAvg can converge to. Our work is different from these works as: 1. We analyze which point the Local-GD can converge to, which is a more elementary problem before obtaining the convergence rate; 2. We use implicit bias as a technical tool to analyze the overparameterized FL.

**Implicit Bias.** Soudry et al. (2018) is the first work to show the gradient descent converges to a max-margin direction on linearly separable data with a linear model and exponentially-tailed loss function. Ji & Telgarsky (2019a) has provided an alternative analysis and extended this to non-separable data. The theory of implicit bias has been further developed, for example, for wide two-layer neural networks Chizat & Bach (2020), deep linear models Ji & Telgarsky (2019b), linear convolutional networks Gunasekar et al. (2018b), two-layer ReLU networks Kou et al. (2024) etc. Beyond gradient descent, more algorithms have been considered, including gradient descent with momentum Gunasekar et al. (2018a), SGD Nacson et al. (2019), Adam Cattaneo et al. (2023), AdamW Xie & Li (2024). Recently, implicit bias has also been used to characterize the dynamics of continual learning, on linear regression Evron et al. (2022); Goldfarb & Hand (2023); Lin et al. (2023), and linear classification Evron et al. (2023). In Evron et al. (2023), gradient descent on continually learned tasks is related to Projections onto Convex Sets (POCS) and shown to converge to a *sequential* max-margin scheme. In our work we consider the implicit bias of gradient descent in distributed setting, which is related to a different parallel projection scheme by projecting onto constraint sets *simultaneously*.

**Parallel Projection.** Parallel projection methods are a family of algorithms to find a common point across multiple constraint sets by projecting onto these sets in parallel. These methods are widely used in feasibility problems in signal processing and image reconstruction Bauschke & Combettes (2011). The straightforward average of multiple projections is known as the simultaneous iterative reconstruction technique (SIRT) in Gilbert (1972). Then de Pierro & Iusem (1984) studied the convergence of PPM for a relaxed version, and Combettes (1994) further generalized the result to inconsistent feasibility problems. In Combettes (1997), an extrapolated parallel projection method was proposed to accelerate the convergence. We note that Jhunjhunwala et al. (2023) used this extrapolation to accelerate FedAvg. However, it was just inspired by the similarity between parallel projection method and FedAvg, while in this work we rigorously prove the relation between PPM and FedAvg using implicit bias of gradient descent.

# B   ADDITIONAL EXPERIMENTS

## B.1   LINEAR CLASSIFICATION WITH DIRICHLET DISTRIBUTION

In federated learning, the Dirichlet distribution is usually used to generate heterogeneous datasets across the compute nodes Hsu et al. (2019); Chen & Chao (2021); Reguieg et al. (2023). For binary classification problem, the Dirichlet distribution $\text{Dir}(\alpha)$ is used to unbalance the positive and negative samples. In the experiments we have 10 compute nodes. We generate 500 samples as $y_i = \text{sign}(x_i^T w^*)$ for $i \in [500]$ and use $\text{Dir}(\alpha)$ to distribute the 500 samples across 10 compute nodes. Note that the number of samples at each compute node is not necessarily identical. Fig. 2 shows performance of Local-GD for linear classification with different parameter $\alpha$ in Dirichlet distribution. The $\lambda$ is set to

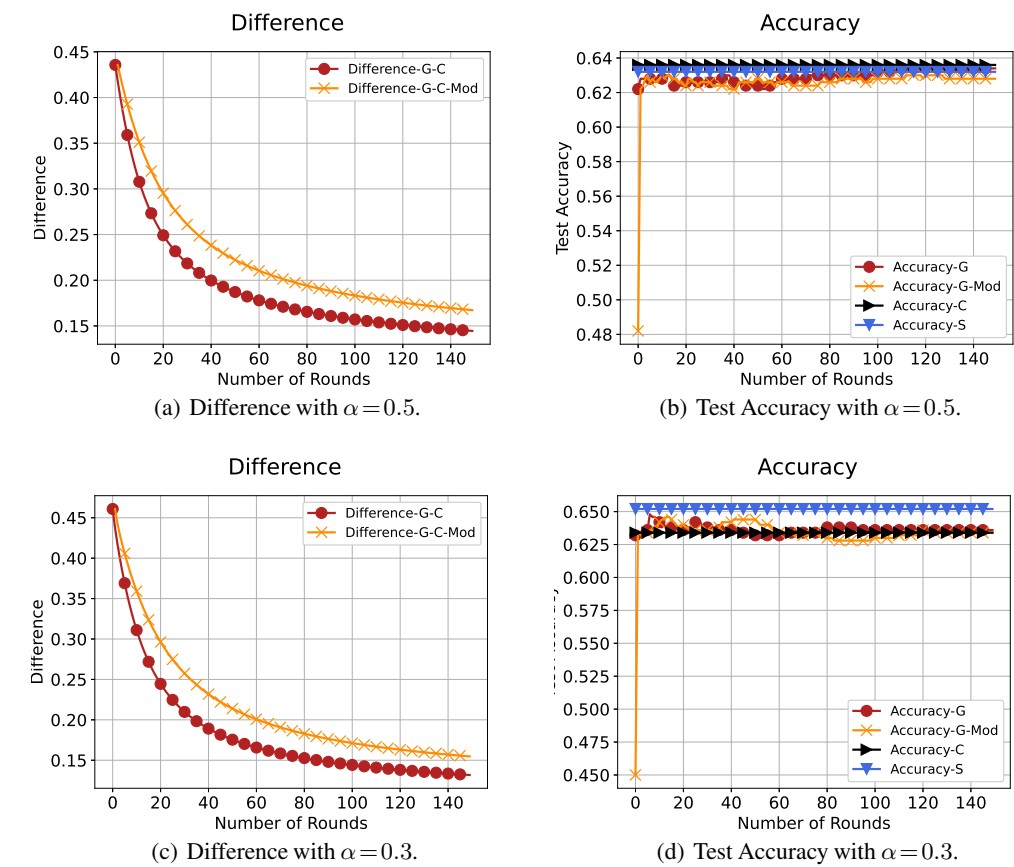

Figure 2: Local-GD on linear classification with Dirichlet distribution.

be 0.0001 and model dimension is fixed as $d = 1500$. The number of local steps $L$ is 150 and number of communication rounds $R$ is 150. The learning rate is 0.01. The centralized model is trained with the same learning rate for 22500 steps. We can see the global model and modified global model still converge to the centralized model in direction and get similar test accuracy.

## C    PROOFS IN SECTION 2

### C.1    PROOF OF LEMMA 1

At each compute node, the local model converges to the solution of problem

$$\min_{w_i} \quad \|w_i - w_0^k\|^2 \quad \text{s.t.} \quad X_i w_i = y_i. \tag{17}$$

Using Lagrange multipliers, we can write the Lagrangian as

$$\frac{1}{2}\|w_i - w_0^k\|^2 + \beta^T(X_i w_i - y_i) \tag{18}$$

Setting the derivative to 0, we know the optimal $\tilde{w}_i$ satisfies

$$\tilde{w}_i - w_0^k + X_i^T \beta = 0, \tag{19}$$

and then

$$\tilde{w}_i = w_0^k - X_i^T \beta. \tag{20}$$

Also by the constraint $y_i = X_i \tilde{w}_i$, we can get

$$y_i = X_i w_0^k - (X_i X_i^T)\beta. \tag{21}$$

Since the model is overparameterized ($d > N$), $X_i X_i^T \in \mathbb{R}^{d \times d}$ is invertible. Then we have

$$\beta = -(X_i X_i^T)^{-1}(y_i - X_i w_0^k). \tag{22}$$

Plugging the $\beta$ back, we can get the closed form solution as

$$\tilde{w}_i = w_0^k + X_i^T (X_i X_i^T)^{-1}(y_i - X_i w_0^k). \tag{23}$$

We update the local model $w_i^{k+1} = \tilde{w}_i$.

We can also write the closed form solution as

$$\begin{aligned} w_i^{k+1} &= w_0^k + X_i^T (X_i X_i^T)^{-1}(y_i - X_i w_0^k) \\ &= \left(I - X_i^T (X_i X_i^T)^{-1} X_i\right) w_0^k + X_i^T (X_i X_i^T)^{-1} y_i \end{aligned} \tag{24}$$

If we plug in the generative model $y_i = X_i w_i^* + z_i$, then the solution is

$$\begin{aligned} w_i^{k+1} &= \left(I - X_i^T (X_i X_i^T)^{-1} X_i\right) w_0^k + X_i^T (X_i X_i^T)^{-1} X_i w_i^* + X_i^T (X_i X_i^T)^{-1} z_i \\ &= (I - P_i) w_0^k + P_i w_i^* + X_i^\dagger z_i. \end{aligned} \tag{25}$$

where $P_i = X_i^T (X_i X_i^T)^{-1} X_i$ is the projection operator to the row space of $X_i$, and $X_i^\dagger = X_i^T (X_i X_i^T)^{-1}$ is the pseudo inverse of $X_i$. It is an interpolation between the initial global model $w_0^k$ and the local true model $w_i^*$, plus a noise term.

After aggregating all the local models, the global model is

$$\begin{aligned} w_0^{k+1} &= \frac{1}{m} \sum_{i=1}^{m} (I - P_i) w_0^k + \frac{1}{m} \sum_{i=1}^{m} P_i w_i^* + \frac{1}{m} \sum_{i=1}^{m} X_i^\dagger z_i \\ &= (I - \bar{P}) w_0^k + \bar{Q} + \bar{Z}, \end{aligned} \tag{26}$$

where $\bar{P} = \frac{1}{m} \sum_{i=1}^{m} P_i, \bar{Q} = \sum_{i=1}^{m} P_i w_i^*, \bar{Z} = \frac{1}{m} \sum_{i=1}^{m} X_i^\dagger z_i$.

After $K$ rounds of communication, the global model is

$$w_0^K = (I - \bar{P})^K w_0^0 + \sum_{k=0}^{K-1} (I - \bar{P})(\bar{Q} + \bar{Z}). \tag{27}$$

If we start from $w_0^0 = 0$, then the solution will converge to $\sum_{k=0}^{K-1} (I - \bar{P})(\bar{Q} + \bar{Z})$.

## C.2  Proof of Theorem 1

We know the difference between global model and centralized model is iteratively projected onto the null space of span of row spaces of $X_i$s:

$$w_0^{k+1} - w_c = (I - \bar{P})(w_0^k - w_c). \tag{28}$$

We can formally describe it as follows. Since the problem is overparameterized globally, we can assume each $X_i$ has full rank $N$. We apply singular value decomposition (SVD) to $X_i$ as $X_i = U_i \Sigma_i V_i^T$, where $U_i \in \mathbb{R}^{N \times N}, V_i \in \mathbb{R}^{d \times N}$. Then $P_i = X_i^T (X_i X_i^T)^{-1} X_i = V_i V_i^T$, which is the projection matrix to the row space of $X_i$.

We apply eigenvalue decomposition on $\bar{P}$ to get $\bar{P} = Q \Sigma Q^T$, where $Q \in \mathbb{R}^{d \times n'}$ and $n'$ is the rank of $\bar{P}$. It satisfies $N \leq n' \leq MN$. Since $\bar{P}$ is a linear combination of $P_i$s, the space of column space of $Q$ is the space spanned by all the vectors $v_{ij}, i = 1, ..., M, j = 1, ..., N$.

We also construct a matrix $Q' \in \mathbb{R}^{d \times (d - n')}$, which consists of orthonomal vectors perpendicular to $Q$. We can project the difference onto column space of $Q$ and $Q'$ respectively.

$$\begin{aligned} Q^T (w_0^{k+1} - w_c) &= Q^T (I - Q \Sigma Q^T)(w_0^k - w_c) = (I - \Sigma) Q^T (w_0^k - w_c) \\ Q'^T (w_0^{k+1} - w_c) &= Q'^T (I - Q \Sigma Q^T)(w_0^k - w_c) = Q'^T (w_0^k - w_c) \end{aligned} \tag{29}$$

After $K$ rounds of communication, we can decomposite $w_0^K - w_c$ into two parts:

$$w_0^K - w_c = QQ^T(w_0^K - w_c) + Q'Q'^T(w_0^K - w_c). \tag{30}$$

Then we can obtain

$$
\begin{aligned}
w_0^K - w_c &= QQ^T(w_0^K - w_c) + Q'Q'^T(w_0^K - w_c) \\
&= Q(I - \Sigma)^K Q^T(w_0^0 - w_c) + Q'Q'^T(w_0^0 - w_c).
\end{aligned}
$$

It shows the initial difference on the column space of $Q$ continues to decrease until zero if $K$ is sufficiently large. And the initial difference on the null space of $Q$ remains constant.

To show the difference $w_0^K - w_c$ goes to zero entirely, we just need to choose an initial point such that initial difference is on the column space of $Q$. When we choose $w_0^0 = 0$, the initial difference is $w_c$ itself. Moreover, the centralized solution $w_c = X_c^T(X_c X_c^T)^{-1} y_c$ exactly lies in the data space spanned by vectors $\{v_{ij}\}_{i=1,j=1}^{M,N}$ since it is a linear combination of columns of $X_c^T$. So if we start from $w_0^0 = 0$, then $w_0^K - w_c$ will go to zero when $K$ is sufficiently large.

# D  PROOFS IN SECTION 3

In the proofs of linear classification, for ease of notation, we redefine the samples $y_{ij}x_{ij}$ to $x_{ij}$ to subsume the labels.

## D.1  PROOFS OF LEMMA 2

We assume $\|w_0^k - \ln(\frac{1}{\lambda})\bar{w}_0^k\| = O(k\ln\ln\frac{1}{\lambda})$. In this case, since $\ln\frac{1}{\lambda}$ grows faster, when $\lambda \to 0$, we can have $\lim_{\lambda \to 0} \frac{w_0^k}{\|w_0^k\|} = \frac{\bar{w}_0^k}{\|\bar{w}_0^k\|}$ for any $k$ at order $o\left(\frac{\ln(1/\lambda)}{\ln\ln(1/\lambda)}\right)$. We will prove it by induction. We define global and local residuals as $r^k = w_0^k - \ln(\frac{1}{\lambda})\bar{w}_0^k$ and $r_i^k = w_i^k - \ln(\frac{1}{\lambda})\bar{w}_i^k$.

When $k = 0$, since $w_0^0 = \bar{w}_0^0 = 0$, $r_i^0 = 0$ and the assumption trivially holds.

When $k \geq 1$, we have

$$
\begin{aligned}
\|r^k\| &= \left\| w_0^k - \ln(\frac{1}{\lambda})\bar{w}_0^k \right\| = \frac{1}{M}\left\| \sum_{i=1}^M w_i^k - \ln(\frac{1}{\lambda})\bar{w}_i^k \right\| \\
&\leq \frac{1}{M}\sum_{i=1}^M \left\| w_i^k - \ln(\frac{1}{\lambda})\bar{w}_i^k \right\| = \frac{1}{M}\sum_{i=1}^M \|r_i^k\|.
\end{aligned}
\tag{31}
$$

where the inequality is triangle inequality. We then focus on the local residual $r_i^k$. We choose an $O(1)$ vector $\tilde{w}_i^k$ and a sign $s_i^k \in \{-1, +1\}$ to show

$$
\begin{aligned}
\|r_i^k\| &= \left\| w_i^k - \left[ \left( \ln(\frac{1}{\lambda}) + s_i^k \ln\ln(\frac{1}{\lambda}) \right)\bar{w}_i^k + \tilde{w}_i^k \right] + s_i^k \ln\ln(\frac{1}{\lambda})\bar{w}_i^k + \tilde{w}_i^k \right\| \\
&\leq \left\| w_i^k - \left[ \left( \ln(\frac{1}{\lambda}) + s_i^k \ln\ln(\frac{1}{\lambda}) \right)\bar{w}_i^k + \tilde{w}_i^k \right] \right\| + \ln\ln(\frac{1}{\lambda})\|\bar{w}_i^k\| + \|\tilde{w}_i^k\|
\end{aligned}
\tag{32}
$$

Recall the $w_i^k$ is the solution of optimization problem

$$\operatorname*{argmin}_{w_i} f_i(w_i) = \sum_{j=1}^N \exp\left(-x_{ij}^T w_i\right) + \frac{\lambda}{2}\|w_i - w_0^{k-1}\|^2, \tag{33}$$

and the loss function $f_i(w_i)$ is a $\lambda$-strongly convex function. Thus we have

$$\|w_i^k - w\| \leq \frac{1}{\lambda}\|\nabla f_i(w)\|, \quad \text{for any } w. \tag{34}$$

Then back to 32, we have

$$\|r_i^k\| \le \underbrace{\frac{1}{\lambda}\left\|\nabla f_i\left[\left(\ln(\frac{1}{\lambda})+s_i^k\ln\ln(\frac{1}{\lambda})\right)\bar{w}_i^k+\tilde{w}_i^k\right]\right\|}_{\|A_i\|}+\ln\ln(\frac{1}{\lambda})\|\bar{w}_i^k\|+\|\tilde{w}_i^k\|. \tag{35}$$

Next we need to show the first term $A_i$ is at $O((k-1)\ln\ln(\frac{1}{\lambda}))$, and also since $\|\bar{w}_i^k\|$ and $\|\tilde{w}_i^k\|$ are $O(1)$ vectors, then $\|r_i^k\|$ is at order $O(k\ln\ln(\frac{1}{\lambda}))$. After averaging, $\|r^k\|$ is also at order $O(k\ln\ln(\frac{1}{\lambda}))$. This confirms the assumption made for induction.

Now we focus on the term $A_i$. The gradient of function $f_i(w)$ is

$$\nabla f_i(w_i)=\sum_j -x_{ij}\exp(-x_{ij}^T w_i)+\lambda(w_i-w_0^{k-1}). \tag{36}$$

The term $A_i$ is

$$A_i=\frac{1}{\lambda}\nabla f_i\left[\left(\ln(\frac{1}{\lambda})+s_i^k\ln\ln(\frac{1}{\lambda})\right)\bar{w}_i^k+\tilde{w}_i^k\right]$$

$$=-\frac{1}{\lambda}\sum_j x_{ij}\exp\left(x_{ij}^T\ln\left(\lambda\ln^{-s_i^k}(\frac{1}{\lambda})\right)\bar{w}_i^k\right)\exp(-x_{ij}^T\tilde{w}_i^k)+\left(\ln(\frac{1}{\lambda})+s_i^k\ln\ln(\frac{1}{\lambda})\right)\bar{w}_i^k+\tilde{w}_i^k-w_0^{k-1}$$

$$=-\frac{1}{\lambda}\sum_j x_{ij}\left(\lambda\ln^{-s_i^k}(\frac{1}{\lambda})\right)^{x_{ij}^T\bar{w}_i^k}\exp(-x_{ij}^T\tilde{w}_i^k)+\left(\ln(\frac{1}{\lambda})+s_i^k\ln\ln(\frac{1}{\lambda})\right)\bar{w}_i^k+\tilde{w}_i^k-w_0^{k-1}. \tag{37}$$

Then we define the set of support vectors as $S_i^k=\{x_{ij}|x_{ij}^T\bar{w}_i^k=1\}$. Recall that we assume $r^{k-1}=w_0^{k-1}-\ln(\frac{1}{\lambda})\bar{w}_0^{k-1}$ is at order $O((k-1)\ln\ln(\frac{1}{\lambda}))$. We can obtain

$$A_i=-\frac{1}{\lambda}\left(\lambda\ln^{-s_i^k}(\frac{1}{\lambda})\right)^1\sum_{x_{ij}\in S_i^k}x_{ij}\exp(-x_{ij}^T\tilde{w}_i^k)-\frac{1}{\lambda}\sum_{x_{ij}\notin S_i^k}x_{ij}\left(\lambda\ln^{-s_i^k}(\frac{1}{\lambda})\right)^{x_{ij}^T\bar{w}_i^k}\exp(-x_{ij}^T\tilde{w}_i^k)$$

$$+\ln(\frac{1}{\lambda})(\bar{w}_i^k-\bar{w}_0^{k-1})-r^{k-1}+s_i^k\ln\ln(\frac{1}{\lambda})\bar{w}_i^k+\tilde{w}_i^k$$

$$=-\ln^{-s_i^k}(\frac{1}{\lambda})\sum_{x_{ij}\in S_i^k}x_{ij}\exp(-x_{ij}^T\tilde{w}_i^k)-\sum_{x_{ij}\notin S_i^k}x_{ij}\lambda^{x_{ij}^T\bar{w}_i^k-1}\left(\ln(\frac{1}{\lambda})\right)^{-s_i^k x_{ij}^T\bar{w}_i^k}\exp(-x_{ij}^T\tilde{w}_i^k)$$

$$+\ln(\frac{1}{\lambda})(\bar{w}_i^k-\bar{w}_0^{k-1})-r^{k-1}+s_i^k\ln\ln(\frac{1}{\lambda})\bar{w}_i^k+\tilde{w}_i^k. \tag{38}$$

By the triangle inequality, we have

$$\|A_i\|\le\underbrace{\left\|\ln(\frac{1}{\lambda})(\bar{w}_i^k-\bar{w}_0^{k-1})-\ln^{-s_i^k}(\frac{1}{\lambda})\sum_{x_{ij}\in S_i^k}x_{ij}\exp(-x_{ij}^T\tilde{w}_i^k)\right\|}_{B_1}$$

$$+\underbrace{\left\|\sum_{x_{ij}\notin S_i^k}x_{ij}\lambda^{x_{ij}^T\bar{w}_i^k-1}\left(\ln(\frac{1}{\lambda})\right)^{-s_i^k x_{ij}^T\bar{w}_i^k}\exp(-x_{ij}^T\tilde{w}_i^k)\right\|}_{B_2}$$

$$+\underbrace{\|r^{k-1}\|}_{O((k-1)\ln\ln(\frac{1}{\lambda}))}+\ln\ln(\frac{1}{\lambda})\underbrace{\|\bar{w}_i^k\|}_{O(1)}+\underbrace{\|\tilde{w}_i^k\|}_{O(1)}. \tag{39}$$

We just need to show $B_1$ and $B_2$ approach to 0 then $\|A_i\|$ can approach to $O(k\ln\ln(\frac{1}{\lambda}))$.

We divide it into two cases.

1. When $\bar{w}_i^k = P(\bar{w}_0^{k-1}) \neq \bar{w}_0^{k-1}$, meaning $\bar{w}_0^{k-1}$ is not in the convex set $C_i$. In this case we choose $s_i^k = -1$ then

$$B_1 = \left\| \ln(\frac{1}{\lambda})(\bar{w}_i^k - \bar{w}_0^{k-1}) - \ln(\frac{1}{\lambda}) \sum_{x_{ij} \in S_i^k} x_{ij} \exp(-x_{ij}^T \tilde{w}_i^k) \right\|$$

$$= \ln(\frac{1}{\lambda}) \left\| (\bar{w}_i^k - \bar{w}_0^{k-1}) - \sum_{x_{ij} \in S_i^k} x_{ij} \exp(-x_{ij}^T \tilde{w}_i^k) \right\|. \tag{40}$$

We now want to choose $\tilde{w}_i^k$ to make $B_1$ as 0. Since $\bar{w}_i^k$ is the solution of SVM problem (13), by the KKT condition of SVM problem, it can be written as

$$\bar{w}_i^k = \bar{w}_0^{k-1} + \sum_{x_{ij} \in S_i^k} \beta_{ij} x_{ij} \tag{41}$$

where $\beta_{ij}$ is the dual varible corresponding to $x_{ij}$ in the set of support vectors. Thus we want to choose $\tilde{w}_i^k$ as

$$\sum_{x_{ij} \in S_i^k} \exp(-x_{ij}^T \tilde{w}_i^k) x_{ij} = \sum_{x_{ij} \in S_i^k} \beta_{ij} x_{ij}. \tag{42}$$

We can prove such a $\tilde{w}_i^k$ almost surely exists in Lemma 5.

For the term $B_2$, since $\lim_{\lambda \to 0} \lambda^{c-1} \ln^c(\frac{1}{\lambda}) \to 0$ for any constant $c > 1$, and $x_{ij}^T \bar{w}_i^k - 1 > 0$ for any $x_{ij}$ being not a support vector, then we can see

$$B_2 = \left\| \sum_{x_{ij} \notin S_i^k} x_{ij} \lambda^{x_{ij}^T \bar{w}_i^k - 1} \left( \ln(\frac{1}{\lambda}) \right)^{x_{ij}^T \bar{w}_i^k} \exp(-x_{ij}^T \tilde{w}_i^k) \right\| \xrightarrow{\lambda \to 0} 0. \tag{43}$$

Here we choose $\tilde{w}_i^k$ and $s_i^k$ to make $B_1 = 0$ and $B_2 \to 0$.

2. When $\bar{w}_i^k = P(\bar{w}_0^{k-1}) = \bar{w}_0^{k-1}$, meaning $\bar{w}_0^{k-1}$ is already in the convex set $C_i$. Then $\bar{w}_i^k - \bar{w}_0^{k-1} = 0$. In this case we choose $\tilde{w}_i^k = 0$ and $s_i^k = +1$. We can have

$$B_1 = \ln^{-1}(\frac{1}{\lambda}) \left\| \sum_{x_{ij} \in S_i^k} x_{ij} \right\| \xrightarrow{\lambda \to 0}, \tag{44}$$

since $\ln^{-1}(\frac{1}{\lambda}) \xrightarrow{\lambda \to 0} 0$ and $\left\| \sum_{x_{ij} \in S_i^k} x_{ij} \right\|$ is $O(1)$.

And since $x_{ij}^T \bar{w}_i^k - 1 > 0$ for any $x_{ij}$ being not a support vector, we have

$$B_2 = \left\| \sum_{x_{ij} \notin S_i^k} x_{ij} \lambda^{x_{ij}^T \bar{w}_i^k - 1} \left( \ln(\frac{1}{\lambda}) \right)^{-x_{ij}^T \bar{w}_i^k} \right\| \xrightarrow{\lambda \to 0} 0, \tag{45}$$

where $\lambda^{x_{ij}^T \bar{w}_i^k - 1} \xrightarrow{\lambda \to 0} 0$ and $\left( \ln(\frac{1}{\lambda}) \right)^{-x_{ij}^T \bar{w}_i^k} \xrightarrow{\lambda \to 0} 0$. Thus we choose $\tilde{w}_i^k$ and $s_i^k$ to make $B_1 \to 0$ and $B_2 \to 0$.

Plugging 39 back into 35, we can obtain

$$\|r_i^k\| \leq \|A_i^k\| + \ln\ln(\frac{1}{\lambda}) \|\bar{w}_i^k\| + \|\tilde{w}_i^k\|$$

$$\leq \underbrace{B_1 + B_2}_{\to 0} + 2\ln\ln(\frac{1}{\lambda}) \|\tilde{w}_i^k\| + 2\|\tilde{w}_i^k\| + \|r^{k-1}\|$$

$$\leq 2\ln\ln(\frac{1}{\lambda}) \|\bar{w}_i^k\| + 2\|\tilde{w}_i^k\| + \|r^{k-1}\|. \tag{46}$$

By the assumption $\|r^{k-1}\| = O((k-1)\ln\ln(\frac{1}{\lambda}))$ and $\|\bar{w}_i^k\| = O(1)$, $\|\tilde{w}_i^k\| = O(1)$, we have $\|r_i^k\| = O(k\ln\ln(\frac{1}{\lambda}))$.

From 31, we finally obtain

$$\|r^k\| \leq \frac{1}{M}\|r_i^k\| = O(k\ln\ln(\frac{1}{\lambda})), \tag{47}$$

which confirms our assumption. Then we have $\lim_{\lambda \to 0} \frac{w_0^k}{\|w_0^k\|} = \frac{\bar{w}_0^k}{\|\bar{w}_0^k\|}$ for any $k$ at order $o\left(\frac{\ln(1/\lambda)}{\ln\ln(1/\lambda)}\right)$.

### D.2  PROOFS OF AUXILIARY LEMMAS

**Lemma 5.** *For the sequence $\{\bar{w}_0^k\}$ generated by sequential SVM problems 13 and aggregations, and for almost all datasets sampled from $M$ continuous distributions, the unique dual solution $\beta_i^k \in \mathbb{R}^{|S_i|\times 1}$ satisfying the KKT conditions of SVM problem 13 has non-zero elements. Then there exists $\tilde{w}_i^k$ satisfying $X_{S_i}\tilde{w}_i^k = -\ln\beta_i^k$.*

For almost all datasets, a hyperplane can be determined by $d$ points. Thus there are at most $d$ support vectors and the set of support vectors is linearly independent.

*Proof.* By the KKT condition of SVM problem, we can write the solution as

$$\bar{w}_i^k = \bar{w}_0^{k-1} + \sum_{x_{ij}\in S_i}\beta_{ij}^k x_{ij} = \bar{w}_0^{k-1} + X_{S_i}^T\beta_i^k. \tag{48}$$

where $X_{S_i} \in \mathbb{R}^{|S_i|\times d}$ is the data matrix with all the support vectors, and $\beta_i^k \in \mathbb{R}^{|S_i|\times 1}$ is the dual variable vector. Thus we can obtain

$$\beta_i^k = (X_{S_i}X_{S_i}^T)^{-1}X_{S_i}(\bar{w}_i^k - \bar{w}_0^{k-1}) = (X_{S_i}X_{S_i}^T)^{-1}\mathbf{1}_{S_i} - (X_{S_i}X_{S_i}^T)^{-1}X_{S_i}\bar{w}_0^{k-1}, \tag{49}$$

where $X_{S_i}X_{S_i}^T$ is invertible since $X_{S_i}$ has full row rank $|S_i|$, and the second equality is from $X_{S_i}\bar{w}_i^k = \mathbf{1}_{S_i}$ with $\mathbf{1}_{S_i} \in \mathbb{R}^{|S_i|\times 1}$ being all one vector. Plugging $\beta_i^k$ back, we have

$$\bar{w}_i^k = \left[I - X_{S_i}^T(X_{S_i}X_{S_i}^T)^{-1}X_{S_i}\right]\bar{w}_0^{k-1} + X_{S_i}^T(X_{S_i}X_{S_i}^T)^{-1}\mathbf{1}_{S_i}. \tag{50}$$

After averaging, the global model is

$$\bar{w}_0^k = \left[I - \frac{1}{M}\sum_{i=1}^M X_{S_i}^T(X_{S_i}X_{S_i}^T)^{-1}X_{S_i}\right]\bar{w}_0^{k-1} + \frac{1}{M}\sum_{i=1}^M X_{S_i}^T(X_{S_i}X_{S_i}^T)^{-1}\mathbf{1}_{S_i}. \tag{51}$$

It implies $\bar{w}_0^k$ is a rational function in the components of $X_1, X_2, \dots, X_M$, and also $\beta_i^k$ is also a rational function in the components of data matrices. So its entries can be expressed as $\beta_{ij}^k = p_{ij}^k(X_1, X_2, \dots, X_M)/q_{ij}^k(X_1, X_2, \dots, X_M)$ for some polynomials $p_{ij}^k, q_{ij}^k$. Note that $\beta_{ij}^k = 0$ only if $p_{ij}^k(X_1, X_2, \dots, X_M) = 0$, and the components of $X_1, X_2, \dots, X_M$ must constitute a root of polynomial $p_{ij}^k$. However, the root of any polynomial has measure zero, unless the polynomial is the zero polynomial, i.e., $p_{ij}^k(X_1, X_2, \dots, X_M) = 0$ for any $X_1, X_2, \dots, X_M$.

Next we need to show $p_{ij}^k$ cannot be zero polynomials. To do this, we just need to construct a specific $X_1, X_2, \dots, X_M$ where the $p_{ij}^k$ is not zero polynomial. Denote $e_i \in \mathbb{R}^d$ as the $i$-th standard unit vector, and $v_1, v_2, \dots, v_M$ be the number of support vectors at $M$ compute nodes. We construct the datasets as

$$X_i = r_i[e_1, e_2, \dots, e_{v_i}]^T, \text{ for all } i. \tag{52}$$

where $r_i$ are positive constants that will be chosen later. For these datasets, the set of support vector is dataset itself, i.e., $X_{S_i} = X_i$. We can calculate

$$X_iX_i^T = r_i^2 I_{v_i}, \ X_i^TX_i = r_i^2\begin{bmatrix} I_{v_i} & \mathbf{0} \\ \mathbf{0} & \mathbf{0}_{(d-v_i)\times(d-v_i)} \end{bmatrix}, \ X_i^T\mathbf{1}_{S_i} = r_i\begin{bmatrix} \mathbf{1}_{v_i} \\ \mathbf{0}_{d-v_i} \end{bmatrix} \tag{53}$$

Thus we have

$$\bar{w}_i^k = \left( I_d - \begin{bmatrix} I_{v_i} & \mathbf{0} \\ \mathbf{0} & \mathbf{0}_{(d-v_i)\times(d-v_i)} \end{bmatrix} \right) \bar{w}_0^{k-1} + \frac{1}{r_i} \begin{bmatrix} \mathbf{1}_{v_i} \\ \mathbf{0}_{d-v_i} \end{bmatrix}. \tag{54}$$

After averaging, the global model in 51 becomes

$$\bar{w}_0^k = \underbrace{\begin{bmatrix} 0 & & & & & & & & \\ & \ddots & & & & & & & \\ & & 0 & & & & & & \\ & & & a_1 & & & & & \\ & & & & \ddots & & & & \\ & & & & & a_{v_{\max}-v_{\min}} & & & \\ & & & & & & 1 & & \\ & & & & & & & \ddots & \\ & & & & & & & & 1 \end{bmatrix}}_{A} \bar{w}_0^{k-1} + \underbrace{\begin{bmatrix} b_1 \\ \vdots \\ b_{v_{\max}} \\ \mathbf{0}_{d-v_{\max}} \end{bmatrix}}_{b}. \tag{55}$$

where $a_j \in \{\frac{1}{M},\frac{2}{M},...,\frac{M-1}{M}\}$ is a constant in the range (0,1), $b_j = \frac{1}{M}\sum_{i\in B_j}\frac{1}{r_i}$ is a positive constant and $B_j \in [M]$ is a set consisting of some compute nodes. Note that $A$ and $b$ are fixed in the iterations and $A$ is a diagonal matrix.

By recursively applying $\bar{w}_0^k = A\bar{w}_0^{k-1} + b$, due to $\bar{w}_0^0 = 0$, we can obtain

$$\bar{w}_0^k = \left( I + A + A^2 + \cdots + A^{k-1} \right) b. \tag{56}$$

Since $A$ is diagonal, the summation is

$$\sum_{j=0}^{k-1} A^j = \begin{bmatrix} 1 & & & & & & & & \\ & \ddots & & & & & & & \\ & & 1 & & & & & & \\ & & & \sum_{j=0}^{k-1} a_1^j & & & & & \\ & & & & \ddots & & & & \\ & & & & & \sum_{j=0}^{k-1} a_{v_{\max}-v_{\min}}^j & & & \\ & & & & & & k & & \\ & & & & & & & \ddots & \\ & & & & & & & & k \end{bmatrix} \tag{57}$$

Recall that

$$\beta_i^k = \left( X_i X_i^T \right)^{-1} \mathbf{1}_{v_i} - \left( X_i X_i^T \right)^{-1} X_i \bar{w}_0^{k-1}$$
$$= \frac{1}{r_i^2} \mathbf{1}_{v_i} - \frac{1}{r_i^2} (\bar{w}_0^{k-1})_{v_i} = \frac{1}{r_i^2} \left( \mathbf{1}_{v_i} - (\bar{w}_0^{k-1})_{v_i} \right). \tag{58}$$

where $(\bar{w}_0^{k-1})_{v_i}$ is the vector with first $v_i$ elements of $\bar{w}_0^{k-1}$.

We need every element of $\beta_i^k$ to be positive, so that we require every element of $(\bar{w}_0^{k-1})_{v_i}$ is less than 1. Then it holds for any $i$-th compute node, thus we require every element of $(\bar{w}_0^{k-1})_{v_{\max}}$ is less than 1. Since $\bar{w}_0^{k-1} = \left( \sum_{j=0}^{k-2} A^j \right) b$, the largest value of $(\bar{w}_0^{k-1})_{v_{\max}}$ satisfies

$$(\bar{w}_0^{k-1})_{\text{largest}} \leq \sum_{j=0}^{k-2} \left( \frac{M-1}{M} \right)^j \times \frac{1}{M} \sum_{i=1}^{M} \frac{1}{r_i^2}$$

$$= M \left( 1 - \left( \frac{M-1}{M} \right)^{k-1} \right) * \frac{1}{M} \sum_{i=1}^{M} \frac{1}{r_i^2} \tag{59}$$

because the maximum value of $a_j$ is $\frac{M-1}{M}$ and the maximum value of $b_j$ is $\frac{1}{M}\sum_{i=1}^{M}\frac{1}{r_i^2}$.

Thus we require

$$\sum_{i=1}^{M}\frac{1}{r_i} < \frac{1}{1-\left(\frac{M-1}{M}\right)^{k-1}}. \tag{60}$$

Since $\left(\frac{M-1}{M}\right)^{k-1} \to 0$ when $k \to \infty$, we only require the left-hand side is less than the lower bound of right-hand side:

$$\sum_{i=1}^{M}\frac{1}{r_i} < 1. \tag{61}$$

Therefore we can choose $r_i = M+1$ to make it happen.

Then we can obtain $\beta_{ij}^k > 0$ holds for any support vector $x_{ij}$ and any round $k$. And the $\tilde{w}_i^k$ simply satisfies $X_{S_i}\tilde{w}_i^k = -\ln\beta_i^k$. $\qquad\square$

## E  LEMMA AND PROOFS IN SECTION 4

Here we provide a lemma of Modified Local-GD similar to Lemma 2 of vanilla Local-GD.

**Lemma 6.** *For almost all datasets sampled from a continuous distribution satisfying Assumption 1, we train the global model $w_0$ from Modified Local-GD in Algorithm 3 and $\bar{w}_0$ from Modified PPM. The parameter is chosen as $\alpha^k = 1 - \frac{1}{k+1}$. With initialization $w_0^0 = \bar{w}_0^0 = 0$, we have $w_0^k \to \ln\left(\frac{1}{\lambda}\right)\bar{w}_0^k$, and the residual $\|w_0^k - \ln\left(\frac{1}{\lambda}\right)\bar{w}_0^k\| = O(k\ln\ln\frac{1}{\lambda})$, as $\lambda \to 0$. It implies that at any round $k = o\left(\frac{\ln(1/\lambda)}{\ln\ln(1/\lambda)}\right)$, $w_0^k$ converges in direction to $\bar{w}_0^k$:*

$$\lim_{\lambda \to 0}\frac{w_0^k}{\|w_0^k\|} = \frac{\bar{w}_0^k}{\|\bar{w}_0^k\|}. \tag{62}$$

*Proof.* With initialization $w_0^0 = \bar{w}_0^0 = 0$, the Modified Local-GD is just a scaling of vanilla Local-GD:

$$w_0^{k+1} = \frac{k}{k+1}\frac{1}{M}\sum_{i=1}^{M}w_i^{k+1}. \tag{63}$$

Also, the Modified PPM is a scaling of vanilla PPM: $\bar{w}_0^{k+1} = \frac{k}{k+1}\frac{1}{M}\sum_{i=1}^{M}\bar{w}_i^{k+1}$.

When $k \geq 1$, we can know the residual between Modified Local-GD and Modified PPM is

$$\|r^k\| = \left\|w_0^k - \ln(\frac{1}{\lambda})\bar{w}_0^k\right\| = \frac{k}{k+1}\frac{1}{M}\left\|\sum_{i=1}^{M}w_i^k - \ln(\frac{1}{\lambda})\bar{w}_i^k\right\|$$

$$\leq \frac{1}{M}\sum_{i=1}^{M}\left\|w_i^k - \ln(\frac{1}{\lambda})\bar{w}_i^k\right\| = \frac{1}{M}\sum_{i=1}^{M}\|r_i^k\|. \tag{64}$$

Then we can follow the same process in the proof of Lemma 2 to obtain

$$\|r^k\| \leq \frac{1}{M}\|r_i^k\| = O(k\ln\ln(\frac{1}{\lambda})), \tag{65}$$

As a result we have $\lim_{\lambda \to 0}\frac{w_0^k}{\|w_0^k\|} = \frac{\bar{w}_0^k}{\|\bar{w}_0^k\|}$.

$\qquad\square$

