# OpenReview forum: "Distributed Gradient Descent with Many Local Steps in Overparameterized Models"
_ICLR.cc/2025/Conference — Submitted to ICLR 2025_

### Official Review · Reviewer_Muac · 2024-11-03

**Soundness:** 2
**Presentation:** 3
**Contribution:** 2
**Rating:** 5
**Confidence:** 3

**Summary:**

This paper investigates the performance of Local Gradient Descent (Local-GD) in distributed training environments, particularly focusing on overparameterized models. It discusses the dynamics of Local-GD under conditions of high local iteration counts and heterogeneous data distributions, areas where traditional convergence analyses have been pessimistic about the performance of such approaches. The authors argue that despite these concerns, Local-GD performs well in practice and attribute this to the implicit bias inherent in gradient descent techniques.

**Strengths:**

1. The paper dives deep into the theoretical aspects of Local-GD for overparameterized linear models, providing an analytical framework that helps explain its effectiveness even with a high number of local steps.
2. The theoretical claims are backed by empirical evidence.

**Weaknesses:**

1. Viewing that the complex, non-linear models dominate modern machine learning applications, the focus on overparameterized linear models restrict the relavance of the results.
2. The **LocalUpdate** methods in this paper are based on gradient descent (GD), which may limit the applicability of the algorithms since stochastic gradient descent (SGD) are much more widely used in distributed and federated learning.
3. The theoretical analysis presents only convergence results, while the rates of convergence and sample complexity are unclear.

**Questions:**

1. The **LocalUpdate** methods seem to perform sufficiently many local steps until convergence (line 290). How does the numerical error incurred by each round of **LocalUpdate** affect the outer-loop convergence at the server side?
2. How to set the values of the parameters $L$ and $\eta$ and how do they affect the overall convergence?
3. In Lemma 2, the model $w_0^k$ converges to $\bar{w}_0^k$ *in direction*. Even if $w_0^k$ tends to be in line with $\bar{w}_0^k$, they can be far away from each other. How does this result make sense?
4. Algorithm 1 line 7: Should it be $w_i^{k+1}$?
5. Constraint in Eq. (13): $\bar{w} \rightarrow w$?

---

> ### Author Response · Authors · 2024-11-20
>
> Thanks for your positive comments and feedback! Please kindly find our responses below.
>
> 1. [Linear models] We agree that non-linear models are more popular in real-world applications. But, understanding the linear models is commonly the first step to establishing theoretical foundations.
> Many phenomenon in machine learning, such as implicit bias, or double-descent, are first understood for linear models. Many theoretical analysis focuses on the linear models first since we can get clean (closed-form) results and non-trivial conclusions in this setting. The implicit bias of non-linear models is still an active area and not well understood. The results on some specific non-linear models are not closed-form (see Line 538-539 in Conclusion and references in Related Work), and it will be cumbersome to analyze the aggregation of these models in distributed setting.
> In experiments, we also fine-tuned a pretrained neural network beyond linear regression and classification.
>
> 2.  [Extension to SGD] It is possible to extend the convergence results of our work to stochastic case based on the implicit bias of SGD in linear models [1]. But understanding the behavior of GD is the necessary first step. Extension to local SGD needs refined analysis of implicit bias of SGD, which will be significantly complicated. In experiments, we use SGD to fine-tune the last layer of a pretrained neural network, which yields similar results on the difference between global model and centralized model, leading us to believe that the behavior of SGD will be similar.
>
> 3. [Convergence rate and sample complexity] Firstly, in this paper we are studying a different problem unlike the standard federated learning papers that usually analyze convergence rate of some optimization algorithm.
> We are showing that, even with a large number of local steps, the centralized solution (with all data together) and the distributed solutions are same in the over-parameterized regime, where this should not have to be the case. In the overparameterized setting, there are many points corresponding to zero training loss. Our work characterizes which specific point Local GD will converge to.  Since each local problem is run for a large number of local steps (to infinity) to use the implicit bias results,
> the general convergence rate will not be well-defined in our setting.
>
> Secondly, we cannot define the sample complexity for our problem, as we have no information about the client distributions $\{D_i\}_{i\in [M]}$, apart from the fact that we have sampled $N$ samples from each of them. Even our results (Theorems 1, 2 and 3) depend on only empirical quantities, for instance, the minima of empirical loss of sampled data points $X_c$ and $y_c$. Obtaining sample complexity by appropriate assumptions on client distributions is therefore an interesting direction for future work.
>
> 4. [Numerical error] If the numerical error refers to the error brought by finite number of local steps, our theoretical result do not guarantee that Local GD would converges to a specific point. However, in experiments these do not show up as a factor, so we assume Local GD to be stable to these errors.
>
> 5. [L and $\eta$] The number of local steps should be large enough to ensure local convergence. In experiments, we set it to be large enough that the local model is stable. The implicit bias of gradient descent towards the minimum-norm solution in linear models does not depend strictly on the learning rate. We only need the learning rate to be sufficiently small to ensure convergence. For example, in linear regression the learning rate $\eta$ is smaller than $\frac{2}{\lambda_{\max}(X^T X)}$, where $\lambda_{\max}(X^T X)$ is the largest eigenvalue. As this value is also the smoothness constant of the objective function, GD with learning rate smaller than this converges. The exact value of $\eta$ does not affect the implicit bias and our conclusions.
>
> 6. [Convergence in direction] For binary linear classification, only the direction of trained model matters, since we use $\mathrm{sign}(x^T w)$ to predict the label. You can rescale the model to any magnitude which does not influence the prediction.
>
> 7. [Typos] Thanks for your careful review. We will correct the typos in the revised version.
>
> References
>
> [1] Mor Shpigel Nacson, Nathan Srebro, and Daniel Soudry. Stochastic gradient descent on separable data:
> Exact convergence with a fixed learning rate. In The 22nd International Conference on Artificial
> Intelligence and Statistics, pp. 3051–3059. PMLR, 2019.

---

> ### Author Response · Authors · 2024-11-27
>
> Dear Reviewer Muac,
>
> We appreciate your time in reviewing our work and providing feedback. We would like to know whether our responses have addressed your concerns. We would be happy to provide further clarification.

---

### Official Review · Reviewer_MuCZ · 2024-11-03

**Soundness:** 2
**Presentation:** 3
**Contribution:** 2
**Rating:** 5
**Confidence:** 4

**Summary:**

This paper attempts to provide theoretical explanation on why FedAvg works in practice when learning overparameterized models with many local steps.

**Strengths:**

The paper is well-written with clear motivation and clean structure.

**Weaknesses:**

1. The analysis, theoretical results, and experiments in this paper focus only on linear regression and linear classification. This is a reasonable starting point but is clearly a severe limitation. This limitation should be stated explicitly in the title and abstract.

2. The statement that "Although the existing convergence analysis suggests that with heterogeneous data, FedAvg encounters quick performance degradation as the number of local steps increases, it is shown to work quite well in practice" in the abstract is questionable. In many recent literatures, FedAvg has shown to stagnate early compared to a centralized approach and the difference increases when the number of clients, number of local steps, and data heterogeneity increases, which motivates the development of various schemes (FedLin, FedSVRG, FedVARP, FedMoS, etc) to mitigate client drifts. Of course, these work do not consider linear models and thus may not contradict to the results shown in this paper. However, I suggest the authors should take these results into account and revise the relevant statements. As for the LLM results with FedAvg, could the authors provide references in which FedAvg is compared to a centralized approach in LLM training? Without comparisons to a centralized baseline, I don't think the fact that LLMs can be trained with 500 local steps is supportive to the statement above.

3. The analysis does not seems to take the stochasticity in SGD into account. The authors could consider discussing potential extensions to the SGD case.

4. The heterogeneous data case considered in the experiment is rather naive. The authors should consider reporting the test results for data distributed based on the Dirichlet distribution as considered in, for example, https://arxiv.org/pdf/2309.01275 and https://arxiv.org/pdf/1909.06335 . This approach allows for a description of level of heterogeneity based on the Dirichlet distribution parameters.

**Questions:**

None.

**Details Of Ethics Concerns:**

None.

---

> ### Author Response · Authors · 2024-11-20
>
> Thanks for your feedback. Please kindly find our responses below.
>
> 1. [Linear models] We have stated in the abstract that we analyzed linear models in the paper. We can add the linear models in the title: though many phenomenon in machine learning, such as implicit bias, or double-descent, are first understood for linear models. In experiments, we also fine-tuned a pretrained neural network beyond linear regression and classification.
> Many theoretical analysis focuses on the linear models first since we can get clean (closed-form) results and non-trivial conclusions in this setting.
> The implicit bias of non-linear models is still an active area and not well understood. The results on some specific non-linear models are not closed-form (see Line 538-539 in Conclusion and references in Related Work), and it will be cumbersome to analyze the aggregation of these models in distributed setting.
>
> 2. [Number of large local steps in practice] The statement that FedAvg with a large number of local steps works well in practice is motivated by the recent work on the training of LLM [1,2]. FedAvg with 500 local steps are compared to centralized training baselines (see Figure 2 and Table 2 in [1], Figure 3 and Table 1 in [2]). The results show the FedAvg can obtain the same or slightly better performance than centralized baseline. They also test it on non-i.i.d. data distribution, which shows the FedAvg with a large number of local steps converges slower on non-i.i.d. data distribution but gets the same final generalization as i.i.d data distribution (see Figure 5 in [1]). In the revised version we specify this statement in the distributed training of LLM. Our work provides theoretical insight into why FedAvg (Local-GD) works well in practice (in linear models).
>
> Also, there are other existing works to argue FedAvg converges very well in real-world tasks even with heterogeneous data [3]. It also points out that the negative theoretical results about FedAvg are mysteriously inconsistent with practical observations. Our work is also an attempt to challenge the conventional wisdom that FedAvg degrades a lot with a large number of local steps in heterogeneous setting.
>
> 3. [Extension to SGD] It is possible to extend the convergence results of our work to stochastic case based on the implicit bias of SGD in linear models [4]. But understanding the behavior of GD is the necessary first step. Extension to local SGD needs refined analysis of implicit bias of SGD, which will be significantly complicated. In experiments, we use SGD to fine-tune the last layer of a pretrained neural network, which yields similar results on the difference between global model and centralized model, leading us to believe that the behavior of SGD will be similar.
>
>
> 4. [Dirichlet distribution] We have added experiments on linear classification with Dirichlet distribution. Note that it is a binary classification problem and we use Dirichlet distribution to unbalance the positive and negative samples. Please see Appendix B in the revised version for the results. The experimental results show that global model obtained from Local-GD can converge to the centralized model in direction with large dimension, which is similar to our reported results on the manuscript.
>
> References
>
> [1] Arthur Douillard, Qixuan Feng, Andrei A Rusu, Rachita Chhaparia, Yani Donchev, Adhiguna Kuncoro,
> Marc’Aurelio Ranzato, Arthur Szlam, and Jiajun Shen. Diloco: Distributed low-communication
> training of language models. arXiv preprint arXiv:2311.08105, 2023.
>
> [2] Sami Jaghouar, Jack Min Ong, and Johannes Hagemann. Opendiloco: An open-source framework
> for globally distributed low-communication training. arXiv preprint arXiv:2407.07852, 2024.
>
> [3] Jianyu Wang, Rudrajit Das, Gauri Joshi, Satyen Kale, Zheng Xu, and Tong Zhang. On the unreasonable
> effectiveness of federated averaging with heterogeneous data. Transactions on Machine Learning
> Research, 2024
>
> [4] Mor Shpigel Nacson, Nathan Srebro, and Daniel Soudry. Stochastic gradient descent on separable data:
> Exact convergence with a fixed learning rate. In The 22nd International Conference on Artificial
> Intelligence and Statistics, pp. 3051–3059. PMLR, 2019.

---

> > ### Comment · Reviewer_MuCZ · 2024-11-26
> >
> > Thank you for the response. I have some further comments, questions, and clarifications on points 1 and 2.
> >
> > 1. [Linear models] I agree that linear models provide a good starting point for the analysis as mentioned in my original comment. However, when a paper focuses only on the linear case, it should be made clear in the title and the abstract. This is particularly important when discussing overparameterized models, which are almost always nonlinear in practice. (It is well-known that overparameterizing a linear model leads to issues such as overfitting and ill-conditioning, which seems to me was confirmed in the results of this paper, e.g., Figure 1 (g).)
> >
> > I also do not understand why the network finetuning test considered here is "beyond linear regression and classification" when the only trainable parameters are the ones in the final linear layer. Couldn't it be reformulated as a linear classification problem by redefining the "input feature" to be the input to the final linear layer?
> >
> > 2. [Large number of local steps in practice] The reference [3] given in the response provides a clear hypothesis under which FedAvg performs well with many local steps. It also gives examples on which FedAvg does not perform well (see Section 3.2 of [3]) when the hypothesis does not hold. The hypothesis proposed in [3] is essentially that the local directions cancel each other at the optimum. The hypothesis should be discussed if [3] is used to support the use of FedAvg with many local steps. Also, the relationship between the hypothesis in [3] and the analysis for overparameterized linear models in this paper is worth discussing. It may help explain the results in the heterogeneous data test with Dirichlet distribution.

---

> ### Author Response · Authors · 2024-11-26
> **Response to additional comments**
>
> 1. [Linear Models] We have mentioned in the abstract that we deal with linear models for regression and classification. As the reviewer correctly points out,  network fine-tuning test is a real-world application of theory of linear models by  considering the input to final layer as the "input feature". We will revise our submission to remove the line "beyond linear regression and classification" and emphasize that it is an application of linear regression and classification.
>
> 2. [Large number of local steps in practice] The reference [3] requires the local objective functions to be strongly convex, see Lemma 1 in [3] , which in our case would correspond to the empirical covariance matrices on each client, $X_i X_i^\top$, to be invertible. For our overparameterized case, $d \geq mn$, each covariance matrix is rank-deficient and is thus not invertible. Therefore, the theory developed in [3] on the benefit of local steps cannot be applied to our setting.
> Note that the intuition in [3] about local directions cancelling each other at optimum is not required in the overparameterized case, as long as a centralized solution exists. As for the results in heterogeneous data sets with Dirichlet distribution in Figure 2, the theory in [3] is not applicable due to overparametrization, however, our theoretical results (Theorems 2 and 3) are applicable and explain the results.

---

> > ### Comment · Reviewer_MuCZ · 2024-11-26
> >
> > Thank you for the clarification. I am updating the score. If the paper is accepted, I suggest to (i) add "linear" to the title and (ii) add proper qualifiers (hypothesis in [3] and/or overparameterized regime, etc) to the statement "Although the existing convergence analysis suggests that with heterogeneous data, FedAvg encounters quick performance degradation as the number of local steps increases, it is shown to work quite well in practice" because there are counterexamples provided even in [3].

---

> > > ### Author Response · Authors · 2024-11-27
> > >
> > > Thank you for reconsidering the score. We will change the title to 'Overparameteried Linear Models' and clarify the hypothesis from [3] in our statement if there is a chance. Meanwhile, we would like to clarify that our theory is still applicable to the counterexamples in [3] (for linear models) which client consensus hypothesis cannot address. That is because our theory does not reply on any assumptions about the data distribution. Additionally the overfitting and ill-conditioning of overparameterized linear models are essential premises to develop benign overfitting and implicit bias in recent advances of learning theory.
> > >
> > > Thanks again for your feedback.

---

### Official Review · Reviewer_uEB6 · 2024-11-04

**Soundness:** 4
**Presentation:** 4
**Contribution:** 2
**Rating:** 6
**Confidence:** 2

**Summary:**

This manuscript solves the contradiction between the results that current theoretical research showed that FedAvg with heterogeneous data encounters quick performance degradation as local updates increase and the fact that FedAvg works well in practice under such conditions. The authors illustrate their theory on the examples of linear regression and linear classification problems from a viewpoint of implicit bias, showing how the result obtained by Local-GD converges with respective to the result of the centralized model. Furthermore, experiments are conducted to validate their theoretical findings.

**Strengths:**

- From the perspective of novelty, this is the first work to analyze the implicit bias of the gradient descent in the distributed setting. The authors propose a distributed algorithm with a refined aggregation method, which shows the exact convergence direction to the centralized model.

- For the linear regression task, the paper presents the closed form solution. Although it is hard for the classification task to get the closed form solution, the authors use other techniques to show the results obtained by Local-GD converge in direction to one point in the global feasible set.

- Experiments are not only conducted on linear regression and linear classification tasks, but also perform fine-tuning of pretrained neural networks.

**Weaknesses:**

- Typos: line 887.

- This manuscript doesn't provide an explicit convergence rate. While there exist works on the convergence rate of Local-SGD on overparameterized model "Faster Convergence of Local SGD for Over-Parameterized Models". Can the authors compare their derived theoretical results with this paper?

- With a large amount of data, stochastic methods might be more popular. Can the theory extend to the Local-SGD to reveal the convergence performance with heterogeneous for overparameterized models, will there be similar convergence results for Local-SGD?

**Questions:**

See Weakness.

---

> ### Author Response · Authors · 2024-11-20
>
> Thanks for your positive comments. Please kindly find our responses below.
>
>
> 1. [Convergence Rate] Note that, we are addressing a different problem than convergence rate of a distributed optimization algorithm (unlike most federated learning papers). We are showing that, even with a large number of local steps, the centralized solution (with all data together) and the distributed solutions are $same$ in the over-parameterized regime, where this should not have to be the case.
> The work you suggested and more work as other reviewers mentioned exploit the property of zero training loss to refine the conventional convergence rate of Local SGD, but do not consider which specific point it will converge to. In overparameterized setting, there are many points corresponding to zero training loss. Our work answers this question by analyzing the implicit bias in distributed setting, which is completely different from the existing work on overparameterized FL. We have included the reference provided, along with some others, in the related works section.
>
> 2. [Extension to SGD] It is possible to extend the convergence results of our work to stochastic case based on the implicit bias of SGD in linear models [1]. But understanding the behavior of GD is the necessary first step. Extension to local SGD needs refined analysis of implicit bias of SGD, which will be significantly complicated. In experiments, we use SGD to fine-tune the last layer of a pretrained neural network, which yields similar results on the difference between global model and centralized model, leading us to believe that the behavior of SGD will be similar.
>
> References
>
> [1] Mor Shpigel Nacson, Nathan Srebro, and Daniel Soudry. Stochastic gradient descent on separable data:
> Exact convergence with a fixed learning rate. In The 22nd International Conference on Artificial
> Intelligence and Statistics, pp. 3051–3059. PMLR, 2019.

---

> > ### Comment · Reviewer_uEB6 · 2024-11-25
> >
> > Thanks for your careful response. My concerns have been solved, and I will maintain my score.

---

> > > ### Author Response · Authors · 2024-11-25
> > >
> > > Thanks for your reply and positive comments again.

---

### Official Review · Reviewer_zEMo · 2024-11-06

**Soundness:** 2
**Presentation:** 3
**Contribution:** 2
**Rating:** 3
**Confidence:** 4

**Summary:**

In the paper, the authors analyze the FedAvg algorithm for overparameterized linear regression and classification problems when the local nodes take a large number of local updates before communicating with the central server. The authors model a large number of local updates by assuming that the local nodes exactly solve their corresponding problems. The authors show that for the regression problem, the aggregation of local models and the centralized model converge to the same solution while for the classification problem, the two solutions converge to the same set. Further, the authors propose a different aggregation rule that guarantees that the centralized model converges in direction to the aggregated local models. Finally, the authors corroborate their theoretical findings via a set of numerical experiments.

**Strengths:**

**Strengths.** Here, I list the major strengths of the paper.

- The paper considers an important problem of analyzing the implicit bias of the FedAvg algorithm which is popular for training large-scale models.
- The attempt to characterize the usefulness of a large number of local updates is useful and of interest to the research community.
- The experiments corroborate the presented theoretical results.
- The paper is well-written and easy to follow.

**Weaknesses:**

**Weaknesses.** Although the paper considers an important problem, I have major doubts about the quality of the paper and the presented results. I list my major concerns in the following.

**Major.**
- I do not understand the argument before equation (8) where the authors state that "We replace $y_i$ in the local model (4)...". Can the authors explain how and why they can substitute $X_i w_c = y_i$ in equation (4)? I do not believe that we should be able to do that since equation (4) is defined for the FedAvg algorithm which is a different algorithm and for FedAvg we have $X_i w_i=y_i$ rather than $X_i w_c = y_i$ which holds for the centralized setting. If I am right equations (8) and (9) will not hold and therefore I have major doubts about the presented results.
- The algorithm for the classification problem should not be called FedAvg or local SGD since it is solving a different regularized problem.
- The result of Theorem 2 utilizes Lemma 3, however, I do not believe the assumptions stated in Lemma 3 to hold true. Specifically, Lemma 3 states that one of the feasible sets is bounded, however, none of the feasible sets in equation (13) are bounded. So how do the results stated in Theorem 2 hold?
- Theorem 2 states that the solution of the centralized model and the distributed algorithm will converge to the same set, however, this result does not convey much since the two points can still be far apart.
- I have a question about the setting of heterogeneous data generation, specifically, after equation (2) the authors mention that $w_i^\ast$ can be different across local nodes and these $w_i^\ast$'s model the heterogeneity across the network. However, in my understanding, the goal of FL is to learn a global model under the setting that the data is generated from a global model and the heterogeneity originates from different noise distributions.  Can the authors please comment on this setting?
- The authors have missed a whole line of work on analyzing overparameterized FL algorithms. Please see [1-3], these should be added to the literature. There may be more, please check.

    [1] Huang, B., Li, X., Song, Z., and Yang, X. Fl-NTK: A neural tangent kernel-based framework for federated learning analysis. In International Conference on Machine Learning, pp. 4423–4434. PMLR, 2021.

    [2] Deng, Y. and Mahdavi, M. Local SGD optimizes overparameterized neural networks in polynomial time. arXivpreprint arXiv:2107.10868, 2021.

    [3] Song B, Khanduri P, Zhang X, Yi J, Hong M. Fedavg converges to zero training loss linearly for overparameterized multi-layer neural networks. In International Conference on Machine Learning PMLR, 2023.


**Minor.**
- When the authors mention in the abstract that the solution to the "classification problem converges (in direction) to the same feasible set as the centralized model" is confusing since reading the statement seems that the two solutions converge to the solutions that have the same direction which is not true.
-   In the introduction, when the authors mention "number of local steps $L$ should not be very large, rather vary as $O( \sqrt{T})$ where $T$ is the total number of iterations". They should clarify the setting under which this statement holds. I believe this does not hold for training non-convex functions but only for strongly-convex smooth problems.
- Once Algorithm 2 is stated, there is no need to re-state Algorithms 3 and 4 since the only difference between the three algorithms is the definition of local functions.

**Questions:**

Please see the weaknesses above.

---

> ### Author Response · Authors · 2024-11-20
>
> Thanks for your feedback! Please kindly find our point-to-point responses below.
>
> 1. [Question Regarding Eq. (8)-(9)] The centralized model is trained with the collection of all local datasets. Therefore the centralized solution $w_c$ should satisfy the constraint $X_c w_c = y_c$, where $X_c = [X_1^T,\dots, X_M^T]^T \in \mathbb{R}^{MN \times d}$ and $y_c = [y_1^T,\dots, y_M^T]^T \in \mathbb{R}^{MN \times 1}$ are just concatenations of $\{X_i\}_ {i=1}^M$ and $\{y_i\}_{i=1}^M$.  Thus it is equivalent to satisfy the constraints $X_i w_c = y_i, \forall i\in [M]$.
> We simply substitute $X_i w_c = y_i$ in the first line of (4) to get (8).
>
> 2. [Local-GD] We do not call this algorithm Fed-Avg, or local-SGD. Local GD as an algorithm has been defined in Algorithm 1. We agree that for the classification problem the loss function includes a weakly regularized term, but we still use a gradient descent with local steps for the algorithm.
>
> 3. [Lemma 3] Thanks for your careful review! Yes, the local feasible set is a polyhedral cone consisting of many hyperplanes. It is unbounded since we consider the overparameterized setting ($d \gg N$). It turns out that, for lemma 3 we do not need the  boundedness condition; From Theorem 1 and Proposition 8 in [1], we can directly obtain Lemma 3 without the bounded-ness condition in our paper.
> The requirement is that the local feasible sets are \textbf{closed, convex sets}. Note that the polyhedral cones are closed and convex. Thus Lemma 3 (with refined condition, see revised version) exactly matches our problem in distributed linear classification, and Theorem 2 in our paper also holds.
>
>
> 4. [Theorem 2 does not show exact convergence] Yes, that is why a modified algorithm is proposed (see Theorem 3).
>
> 5. [Data heterogeneity] Your proposed setting is just a special case of our setting. What you suggested can be seen as a special case of (2), where ground truth models $\{w_i^*\}_{i=1}^M$ can be exactly same and serve as the global model. The distribution of noise $z_i$ across compute nodes can be different. This setting is an easier case and all of our results are applicable to this special case.
> In general, there are different kinds of data heterogeneity in federated learning literature, such as label heterogeneity and feature heterogeneity, see Section 2.1 in [2]. In our setting of linear models, the ground truth models that generate local datasets can be different. This is a relatively strong setting as different compute nodes can have different labels even without noise for the same feature $x$.
>
> 6. [Works in Overparameterized FL] Thanks for introducing us to this line of work! We mainly focus on the introduction of implicit bias in distributed setting. We have included these works in our discussion on related works.
> The common point of these works is that they plug in the property that FedAvg can converge to zero training loss into the convergence analysis, and analyze the convergence rate. However, they do not guarantee which point FedAvg can converge to. Our work is different from these works as: 1. We analyze which point the Local-GD can converge to, which is a more elementary problem before obtaining the convergence rate; 2. We  use implicit bias as a technical tool to analyze the overparameterized FL.
>
> 7. [Conditions on the number of local steps] Yes, the conclusion that the number of local steps should be no larger than $O(\sqrt{T})$ applies to strongly convex and smooth loss functions[3,4]. We specify this in the revised version. Note that [3] requires $O(\sqrt{T})$ local steps for iid client data in this setting, while Theorem 1 in [4] shows that if the loss function is also Lipschitz, for non-iid clients the number of local steps must not exceed $\Omega(\sqrt{T})$.
>
> 8. [Algorithm 3 and 4] We have removed Algorithm 3 and 4 in the revised version as you suggested. The linear regression and classification are covered with the same Local-GD (Algorithm 1 and 2) with different loss functions.
>
> References
>
> [1] Patrick L Combettes. Inconsistent signal feasibility problems: Least-squares solutions in a product
> space. IEEE Transactions on Signal Processing, 42(11):2955–2966, 1994.
>
> [2] Mang Ye, Xiuwen Fang, Bo Du, Pong C. Yuen, and Dacheng Tao. 2023. Heterogeneous Federated Learning: State-of-the-art and Research Challenges. ACM Comput. Surv. 56, 3, Article 79.
>
> [3] Sebastian U Stich. Local SGD converges fast and communicates little. In International Conference on Learning Representations, 2019.
>
> [4] Xiang Li, Kaixuan Huang, Wenhao Yang, Shusen Wang, and Zhihua Zhang. On the convergence of FedAvg on non-IID data. In International Conference on Learning Representations, 2020.

---

> ### Author Response · Authors · 2024-11-27
>
> Dear Reviewer zEMo,
>
> We appreciate your time in reviewing our work and providing feedback. We would like to know whether our responses have addressed your concerns. We would be happy to provide further clarification.

---

### Author Response · Authors · 2024-11-20

Thanks for the comments and feedback provided by the reviewers. We have revised the manuscript according to the feedback and answered the questions from reviewers sequentially below. The revised parts are marked with blue color. In summary, our revision is to:

1. Remove Algorithm 3 and 4 to reduce redundancy;

2. Revise Lemma 3 for a more general condition;

3. Add a discussion of a line of work about overparameterized models in federated learning in Related Work;

4. Add experiments in linear classification with Dirichlet distribution in Appendix B;

5. Change some expressions to be more accurate and clear as suggested;

6. Correct the typos.

We would like to note that our contribution is to introduce the analysis of implicit bias in distributed setting in contrast to the normal convergence analysis of Local-GD, and conclude that Local-GD can get similar models as centralized training even with a large number of local steps in heterogeneous setting. Our results are distribution-independent and best seen as learning theoretic, as opposed to convergence analysis of some federated learning algorithm.

---

### Meta-Review · Area_Chair_i7aN · 2024-12-21

**Metareview:**

This paper studies an interesting problem in distributed learning. The authors analyze how Local Gradient Descent (Local-GD) works with many local steps in overparameterized models, especially when data is different across devices. They show theoretical results about the “implicit bias” of Local-GD and explain why it can work well in practice. The paper is well-written and includes experiments to support the theory, including applications to fine-tuning neural networks. The idea of analyzing implicit bias in distributed learning is new and helpful for understanding Local-GD better.

However, reviewers expressed a few concerns with the current version. The main reason is that the study focuses only on linear models, which are too simple compared to the non-linear models used in most real-world machine learning. This makes the results less useful for many practical problems. Also, the paper does not include important information, like how fast the method converges or how to apply it to more complex methods like Local-SGD. Some reviewers felt that experiments were not enough to show that the results work well in real-world settings with diverse data. Some reviewers also pointed out that similar ideas have already been explored in other works, so the contribution is not very strong.

**Additional Comments On Reviewer Discussion:**

During the discussion and rebuttal period, reviewers raised several points about the paper. They questioned the limited focus on linear models and suggested that this made the results less relevant for real-world machine learning problems, which often involve non-linear models. There were also concerns about the lack of convergence rates and the limited exploration of practical scenarios like stochastic methods or more complex heterogeneous data settings. The authors responded by clarifying their focus on linear models as a first step and adding some new experiments, but they did not fully address the concerns about the broader applicability of their findings. While the authors provided detailed explanations for their theoretical results, these were still seen as narrow and not general enough for a wide range of applications.

---

### Decision · Program_Chairs · 2025-01-22

Reject